

# Revised phylogenetic analysis of the Aetosauria (Archosauria: Pseudosuchia); assessing the effects of incongruent morphological character sets

William G. Parker[1,2]

[1] Division of Resource Management, Petrified Forest National Park, Arizona, United States
[2] Jackson School of Geosciences, University of Texas at Austin, Austin, Texas, United States

Corresponding author
William G. Parker,
William_Parker@nps.gov

## ABSTRACT

Aetosauria is an early-diverging clade of pseudosuchians (crocodile-line archosaurs) that had a global distribution and high species diversity as a key component of various Late Triassic terrestrial faunas. It is one of only two Late Triassic clades of large herbivorous archosaurs, and thus served a critical ecological role. Nonetheless, aetosaur phylogenetic relationships are still poorly understood, owing to an overreliance on osteoderm characters, which are often poorly constructed and suspected to be highly homoplastic. A new phylogenetic analysis of the Aetosauria, comprising 27 taxa and 83 characters, includes more than 40 new characters that focus on better sampling the cranial and endoskeletal regions, and represents the most comprenhensive phylogeny of the clade to date. Parsimony analysis recovered three most parsimonious trees; the strict consensus of these trees finds an Aetosauria that is divided into two main clades: Desmatosuchia, which includes the Desmatosuchinae and the Stagonolepidinae, and Aetosaurinae, which includes the Typothoracinae. As defined Desmatosuchinae now contains *Neoaetosauroides engaeus* and several taxa that were previously referred to the genus *Stagonolepis,* and a new clade, Desmatosuchini, is erected for taxa more closely related to *Desmatosuchus.* Overall support for some clades is still weak, and Partitioned Bremer Support (PBS) is applied for the first time to a strictly morphological dataset demonstrating that this weak support is in part because of conflict in the phylogenetic signals of cranial versus postcranial characters. PBS helps identify homoplasy among characters from various body regions, presumably the result of convergent evolution within discrete anatomical modules. It is likely that at least some of this character conflict results from different body regions evolving at different rates, which may have been under different selective pressures.

## INTRODUCTION

The goal of phylogenetic systematics is to determine phylogenetic relationships of organisms based on shared homologous character states, and to use this information to interpret the evolutionary histories of clades, or monophyletic lineages of organisms, as
well as the histories of various evolutionary character transformations (e.g., *Wiley & Lieberman, 2011*). This presents special challenges for vertebrate groups with extensive carapaces of dermal armor like those of aetosaurian and ankylosaurid archosaurs, which are comprised of hundreds of individual osteoderms (e.g., *Desojo et al., 2013*). Whereas these osteoderms may be common in the fossil record, they are generally dissociated from the rest of the skeleton prior to burial (*Heckert & Lucas, 2000*). It has been asserted for aetosaurians that osteoderms provide an exhaustive source of phylogenetically informative character data above and beyond that provided by the underlying skeleton (e.g., *Long & Ballew, 1985*; *Heckert & Lucas, 1999*; *Parker, 2007*), but it has also been argued that, while informative, these data may be plagued with phylogenetically confounding homoplasy (*Parker, 2007*; *Parker, 2008a*). The specific goal of this paper is to confront these assertions analytically, first by undertaking an expanded phylogeny of aetosaurian archosaurs based on the largest taxonomic sample yet assembled, using a suite of characters that samples both osteoderms and endoskeletal characters; and second, by applying a new method (Partitioned Bremer Support) to assess character support and conflict within an entirely morphological dataset.

## Historical background

Aetosaurians are quadrupedal, pseudosuchian archosaurs characterized by antero-posteriorly shortened skulls with upturned snouts, and heavy armor carapaces, as well an armor plastron (Fig. 1; *Walker, 1961*; *Desojo et al., 2013*). They had a global distribution during the Late Triassic and are often used as index fossils for biostratigraphic correlations (*Heckert & Lucas, 1999*; *Desojo et al., 2013*). The paramedian osteoderms possess a diagnostic surface ornamentation that allows for assignment of osteoderms and associated material to specific taxa, although as previously mentioned some of these osteoderm characters may be homoplastic (*Long & Ballew, 1985*; *Parker, 2007*). Accordingly it has been argued that characters from the lateral osteoderms may be more phylogenetically informative than those from the paramedian series (*Parker, 2007*).

When *Long & Ballew (1985)* first proposed a taxonomy of North American aetosaurs based exclusively on osteoderm characters, they recognized only four taxa (*Desmatosuchus*, *Typothorax*, *Calyptosuchus*, *Paratypothorax*). Much new work based upon many new specimens reveals that the particular osteoderm character combinations proposed by *Long & Ballew (1985)* in fact can occur in many other unique combinations, resulting in the establishment of many new taxa from North America based almost solely on osteoderms (e.g., *Zeigler, Heckert & Lucas, 2003*; *Martz & Small, 2006*; *Spielmann et al., 2006*; *Lucas, Hunt & Spielmann, 2007*; *Parker, Stocker & Irmis, 2008*; *Heckert et al., 2015*). Moreover, it has been demonstrated that aetosaurs with nearly identical osteoderm character combinations can differ significantly in the other portions of the skeleton, especially in the cranial elements, indicating even more taxonomic potential (*Desojo, 2005*; *Desojo & Báez, 2005*; *Desojo & Báez, 2007*; *Desojo & Ezcurra, 2011*). Finally, aetosaurian osteoderm characters are not intraorganisimally homogeneous (i.e. characters can differ depending on position within the same carapace) and capturing
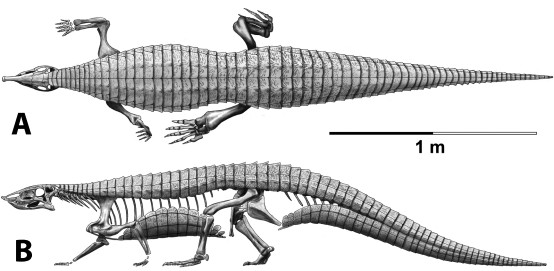

**Figure 1 Skeletal reconstruction of an aetosaur (*Stagonolepis robertsoni*) showing the extensive carapace and associated armor in dorsal (A) and lateral (B) views.** Courtesy of Jeffrey Martz.

this variation in the construction of phylogenetically informative characters is challenging (*Harris, Gower & Wilkinson, 2003*; *Parker, 2007*; *Parker, 2008b*; *Desojo et al., 2013*).

Although early studies did focus on character change across broadly defined carapace regions such as the cervical and caudal regions (e.g., *Long & Ballew, 1985*; *Heckert & Lucas, 1999*), more recent studies have sought to detail variation within those subregions (*Martz, 2002*; *Parker, 2003*; *Parker, 2008b*; *Schoch, 2007*; *Parker & Martz, 2010*; *Heckert et al., 2015*). Potentially further complicating this situation is our general lack of data regarding character transformations affected by ontogenetic variation as well as differences caused by individual and sexual dimorphism (*Taborda, Cerda & Desojo, 2013*; *Taborda, Heckert & Desojo, 2015*). Overall though, the rich source of character data present in aetosaurian osteoderms provides the systematist with a broad canvas on which to construct a detailed phylogenetic hypothesis, presuming of course that the changes in osteoderm characters are indeed phylogenetically informative (*Parker, 2007*) and that the homology of these characters can be determined (e.g., *Harris, Gower & Wilkinson, 2003*; *Parker & Martz, 2010*; *Heckert et al., 2015*).

At present we do not have an appropriate sample size across all clades to capture all of intraorganisimal character variation that occurs across the aetosaurian carapace and plastron. Indeed, many taxa are currently only known from a handful of associated osteoderms (e.g., *Tecovasuchus*, *Apachesuchus*, *Rioarribasuchus*), with the current challenge simply lying in determining the proper position of these osteoderms within the carapace (*Lucas, Heckert & Hunt, 2003*; *Martz & Small, 2006*; *Spielmann et al., 2006*; *Parker, 2007*; *Lucas, Hunt & Spielmann, 2007*; *Parker & Martz, 2010*; *Spielmann & Lucas, 2012*). As more discoveries are made, particularly of associated and articulated specimens, our increased understanding of positional variation should allow for more precise placement of isolated osteoderms leading to stronger determinations of homology of individual osteoderms (*Parker, 2007*; *Parker & Martz, 2010*; *Heckert et al., 2015*).

For this study all previously recommended characters used for determination of aetosaurian systematics were reviewed (*Parrish, 1994*; *Heckert, Hunt & Lucas, 1996*; *Heckert & Lucas, 1999*; *Desojo, 2005*; *Parker, 2007*; *Desojo, Ezcurra & Kischlat, 2012*; *Roberto-Da-Silva et al., 2014*; *Heckert et al., 2015*). Characters were discarded if found to be generally uninformative or ambiguously scored. The retained characters, as well as new characters, have been rewritten to be more descriptive and thus hopefully easier to

interpret and score. Although the retention and construction of many characters and associated character states would presumably lead to better resolution and clade support (*Hillis, Huelsenbeck & Cunningham, 1994*), the goal of any phylogenetic analysis is accuracy, and this should not come at the expense of artificial resolution by including ambiguously written characters (*Slowinski, 1993*). Thus, the overarching goal of this project was to recover phylogenetic trees that seem logical given our anatomical understanding of aetosaurians, rather than highly resolved and supported trees that appear problematic and nonsensical in these regards. The matrix of *Parker (2007)*, which has been used as the basis for many recent phylogenetic analyses (*Parker, Stocker & Irmis, 2008*; *Desojo, Ezcurra & Kischlat, 2012*, *Roberto-Da-Silva et al., 2014*; *Heckert et al., 2015*), is dominated by osteoderm characters. This is problematic given the large amount of discovered homoplasy in this dataset (*Parker, 2007*; *Desojo, Ezcurra & Kischlat, 2012*), and in light of the underlying assumption that osteoderm characters provide the main phylogenetic signal for the clade irrespective of the rest of the skeleton (*Desojo, 2005*; *Parker, 2007*; *Parker, 2008b*). For these reasons, this study sought to increase the number of non-osteoderm characters, as suggested by *Desojo (2005)* & *Desojo, Ezcurra & Kischlat (2012)*. This presents challenges because of the relative infrequency of aetosaurian postcranial remains, which are lacking for many taxa or sometimes obscured by articulated carapaces. One of the best sources for aetosaurian postcranial bones is the *Placerias* Quarry in northeastern Arizona (*Long & Murry, 1995*). However, owing to a lack of association with diagnostic osteoderm material, most of these postcranial elements cannot unequivocally be referred to species (*Parker, 2014*; *Parker, 2005a*; differing from *Long & Murry, 1995*). Fortunately, there is cranial material preserved for many aetosaurian taxa and almost every known skull, with the exception of some elements from the *Placerias* Quarry and the Post Quarry (Texas), are unambiguously associated with osteoderms allowing for a precise taxonomic referral. Thus, the present analysis was able to significantly expand the number of cranial characters utilized.

The original basis for aetosaurian phylogenetic characters and character transformations is a table of information published by *Long & Ballew (1985*:58*)* where comparisons are provided between various North American taxa, establishing a key early character-based taxonomic scheme for aetosaurians (also see *Walker, 1961*). Several of these characters are still utilized in recent phylogenetic analyses. The first computed phylogenetic analysis of aetosaurians (*Parrish, 1994*) examined 15 characters (six osteoderm, nine non-osteoderm) and eight taxa. However, nine of those characters are parsimony-uninformative for the ingroup, and there are several incorrect scorings and typographical errors that affect the analysis; thus the published tree is neither well-resolved, nor accurate in its character state distributions (*Harris, Gower & Wilkinson, 2003*). *Heckert, Hunt & Lucas (1996)* expanded on *Parrish's (1994)* work, inflating the matrix to nine taxa and 22 (potentially 23) characters (17 armor, five non-armor). That study was also affected by some scoring errors, as well as the lack of use of a non-aetosaurian outgroup to root the resulting trees (*Harris, Gower & Wilkinson, 2003*), but did include many new characters that have been used in subsequent

aetosaurian phylogenetic studies. Furthermore that study was the first to unambiguously recover the major clades Desmatosuchinae and Typothoracisinae (*sensu Parker, 2007*).

Heckert & Lucas (1999) aimed to expand the matrix of Heckert, Hunt & Lucas (1996), mainly to determine the phylogenetic relationships of a new taxon, *Coahomasuchus kahleorum*. Their published matrix consisted of 13 in-group taxa and 60 characters. However, 26 of these characters as coded were parsimony uninformative, and as noted by Harris, Gower & Wilkinson (2003) the published matrix included several typographical errors. When corrected, that matrix produced a tree that was different from the published one. Harris, Gower & Wilkinson (2003) were critical of several other aspects of this study, including the ad hoc deletion of taxa from the matrix when safe methods to determine appropriate taxon deletion were available (e.g., Wilkinson, 1995a), and character constructions that inflated seemingly non-independent character suites and biased the resulting tree (composite versus reductive coding; Rowe, 1988; Wilkinson, 1995b). Nonetheless, the study by Heckert & Lucas (1999) built further upon the character list of Heckert, Hunt & Lucas (1996) and represents a very important progression in our understanding of aetosaurian systematics.

The most recent core phylogenetic analysis of aetosaurians (Parker, 2007) focused on the lateral osteoderms, whereas the previous studies had focused more on characters of the paramedian osteoderms (Heckert, Hunt & Lucas, 1996; Heckert & Lucas, 1999). Parker (2007) noted that aetosaurians could roughly be divided into three groups based on the overall anatomy of the lateral osteoderms. This translated into a phylogenetic analysis (16 in-group taxa, 37 characters) that recovered three distinct clades: Aetosaurinae, Desmatosuchinae (Heckert & Lucas, 2000) and Typothoracinae. Whereas support for Desmatosuchinae and Typothoracinae was strong, especially for the subclade Paratypothoracini, Aetosaurinae was unresolved and weakly supported. This became especially apparent when other taxa were subsequently added to the matrix, causing significantly different tree topologies and character support (Parker, Stocker & Irmis, 2008; Desojo, Ezcurra & Kischlat, 2012). Indeed, a recent study (Desojo, Ezcurra & Kischlat, 2012) failed to recover Aetosaurinae as a clade, with *Aetosaurus ferratus* as the only member by definition (Heckert & Lucas, 2000). Desmatosuchinae is always recovered and well-supported, but relationships within the clade are not always fully resolved (e.g., Parker, Stocker & Irmis, 2008); however, Typothoracinae remains well-supported and resolved. Nonetheless, criticisms of the Parker (2007) dataset include the lack of endoskeletal characters as well as some scoring errors (see Desojo & Ezcurra, 2011; Desojo, Ezcurra & Kischlat, 2012; Heckert et al., 2015).

## Materials and Methods

In order to test these questions about taxon sampling, character independence, and tree topology, the matrix has been expanded to include more taxa and characters. The new matrix (Appendix A) utilizes 83 characters for 26 ingroup taxa. The characters are well-divided between anatomical regions, with endoskeletal characters constituting the majority (34 cranial, 16 axial/appendicular, 33 osteoderm).

The 26 in-group taxa include the majority of aetosaurian taxa currently considered valid (*Desojo et al., 2013*; *Roberto-Da-Silva et al., 2014*; *Heckert et al., 2015*). They are listed below, and this study is the first to investigate the phylogenetic positions of *Adamanasuchus eisenhardtae*, *Apachesuchus heckerti*, *Stagonolepis olenkae*, *Redondasuchus rineharti* as well as a new taxon, *Scutarx deltatylus gen. et sp. nov.* Other taxa are rescored (e.g., *Coahomasuchus kahleorum*; *Typothorax coccinarum*) based on new referred material.

Taxa excluded from this analysis include *Acaenasuchus geoffreyi* (*Long & Murry, 1995*; *Redondasuchus reseri Hunt & Lucas, 1991*; *Typothorax antiquum Lucas, Heckert & Hunt, 2003*; and *Chilenosuchus forttae Casamiquela, 1980*). *Acaenasuchus* and *Chilenosuchus* were excluded because *Chilenosuchus* presently scores as a taxonomic equivalent (*sensu Wilkinson, 1995a*) of *Typothorax coccinarum*, and newly recognized material, including vertebrae and fused osteoderms, of *Acaenasuchus* casts doubt on its aetosaurian identify (M. Smith, personal communication, 2014). *Redondasuchus reseri* is poorly known and presently scores as a taxonomic equivalent of *Redondasuchus rineharti*; whereas *Typothorax antiquum* represents an ontogenetic stage of *Typothorax coccinarum* rather than a distinct species (*Parker, 2006*; *Parker & Martz, 2011*; *Martz et al., 2013*). In any case, in this matrix *Typothorax antiquum* and *Typothorax coccinarum* are taxonomic equivalents (i.e., they are scored exactly the same, and thus can obscure relationships in the data if both are included; *Wilkinson, 1995a*), so the less complete, *Typothorax antiquum*, is excluded.

*Revueltosaurus callenderi* is included in the analysis as an outgroup because it is currently recovered as the sister taxon of Aetosauria (*Nesbitt, 2011*). Furthermore, it is known from several specimens, which preserve nearly the entire skeleton (*Parker et al., 2007*). *Postosuchus kirkpatricki* is utilized as an outgroup because it is relatively complete, well-described and illustrated (*Weinbaum, 2011*; *Weinbaum, 2013*). Furthermore, it represents a more crownward clade (Paracrocodylomorpha) within Pseudosuchia providing a deeper optimization of character states than can be provided by *Revueltosaurus*. Both of these taxa have been utilized as outgroups in previous phylogenetic studies of the Aetosauria (e.g., *Heckert & Lucas, 1999*; *Parker, 2007*; *Desojo, Ezcurra & Kischlat, 2012*; *Heckert et al., 2015*). Unfortunately neither *Postosuchus* nor *Revueltosaurus* can presently be scored for lateral osteoderm characters and therefore these characters have been scored as inapplicable for these taxa. Furthermore, most of the paramedian osteoderm characters were scored as inapplicable for *Postosuchus* because even though *Postosuchus* possesses trunk osteoderms, the homology of characters such as ornamentation pattern and presence of certain processes cannot be determined.

A previous work (*Parker, 2007*) incorporated many scorings from past studies (*Parrish, 1994*; *Heckert, Hunt & Lucas, 1996*; *Heckert & Lucas, 1999*) some of which were later determined to be erroneous (*Schoch, 2007*; *Desojo & Ezcurra, 2011*; *Desojo, Ezcurra & Kischlat, 2012*; *Heckert et al., 2015*). Therefore, for this study the matrix was scored from scratch and the scorings completed from carefully studying materials first hand for most taxa, and using photos and the literature for any not studied first-hand (*Stagonolepis olenkae*, *Aetosaurus ferratus*, SMNS 19003 (*Desojo & Schoch, 2014*), *Stenomyti huangae*, *Redondasuchus rineharti*, *Gorgetosuchus pekinensis*, *Polesinesuchus aurelioi*). Much effort

was directed toward detecting and fixing typographic errors, which can have a major effect on the final tree topologies (*Harris, Gower & Wilkinson, 2003*). Scoring completeness is shown in Supplemental Table 1 for each taxon, with inapplicable characters counted as scored. Completeness scores range from 98% (80 of 82) for *Desmatosuchus smalli*, which is known from several skulls and skeletons; to 22% for *Apachesuchus heckerti* (18 of 82), which is known only from five paramedian osteoderms. The average completeness score was 60%. The major factor causing incompleteness is a lack of skull material, which affected all taxa that scored lower than 50%. Because aetosaurians are generally identified by armor characters, there are no taxa that consist solely of cranial material, in contrast with many other groups (e.g., synapsids, dinosaurs).

The electronic version of this article in Portable Document Format (PDF) will represent a published work according to the International Commission on Zoological Nomenclature (ICZN), and hence the new names contained in the electronic version are effectively published under that Code from the electronic edition alone. This published work and the nomenclatural acts it contains have been registered in ZooBank, the online registration system for the ICZN. The ZooBank LSIDs (Life Science Identifiers) can be resolved and the associated information viewed through any standard web browser by appending the LSID to the prefix http://zoobank.org/. The LSID for this publication is: urn:lsid:zoobank.org:pub:841F81C7-A4AE-4146-94FE-DFE0A6725634. The online version of this work is archived and available from the following digital repositories: PeerJ, PubMed Central and CLOCKSS.

*Institutional abbreviations* – **AMNH**, American Museum of Natural History, New York, USA; **ANSP**, Academy of Natural Sciences of Drexel University, Philadelphia, Pennsylvania, USA; **CPE2**, Coleção Municipal, São Pedro do Sul, Brazil; **DMNH**, Perot Museum of Natural History, Dallas, Texas, USA; **DMNH**, Denver Museum of Nature and Science, Denver, Colorado, USA; **FMNH**, Field Museum, Chicago, IL, USA; **FR**, Frick Collection, American Museum of Natural History, New York, USA; **MCCDP**, Mesalands Community College Dinosaur Museum, Tucumcari, New Mexico, USA; **MCSNB,** Museo Civico di Scienze Naturali Bergamo, Bergamo, Italy; **MCP**, Museo de Ciencias e Tecnología, Porto Alegre, Brazil; **MCZ**, Museum of Comparative Zoology, Harvard University, Cambridge, Massachusetts, USA; **MCZD**, Marischal College Zoology Department, University of Aberdeen, Aberdeen, Scotland, UK; **NCSM**, North Carolina State Museum, Raleigh, North Carolina, USA; **NHMUK**, The Natural History Museum, London, United Kingdom; **NMMNH**, New Mexico Museum of Natural History and Science, Albuquerque, New Mexico, USA; **MNA**, Museum of Northern Arizona, Flagstaff, Arizona, USA; **PEFO**, Petrified Forest National Park, Petrified Forest, Arizona, USA; **PFV**, Petrified Forest National Park Vertebrate Locality, Petrified Forest, Arizona, USA; **PVL**, Paleontología de Vertebrados, Instituto 'Miguel Lillo', San Miguel de Tucumán, Argentina; División de Paleontología de Vertebrados del Museo de Ciencias Naturales y Universidad Nacional de San Juan, San Juan, Argentina, **SMNS**, Staatliches Museum für Naturkunde, Stuttgart, Germany; **TMM**, Texas Memorial Museum, Austin, Texas, USA; **TTUP**, Museum of Texas Tech, Lubbock, Texas, USA; **UCMP**, University of California, Berkeley, California, USA; **ULBRA PVT**, Universidade Luterana do Brasil,

Coleção de Paleovertebrados, Canoas, Rio Grande do Sul, Brazil; **UMMP**, University of Michigan, Ann Arbor, Michigan, USA; **USNM**, National Museum of Natural History, Smithsonian Institution, Washington, D.C., USA; **VPL**, Vertebrate Paleontology Lab, University of Texas at Austin, Austin, Texas, USA; **YPM**, Yale University, Peabody Museum of Natural History, New Haven, Connecticut, USA; **VRPH**, Sierra College, Rocklin, California, USA; **ZPAL,** Institute of Paleobiology of the Polish Academy of Sciences in Warsaw, Warsaw; Poland.

## TERMINAL TAXA

The phylogenetic study by *Nesbitt (2011)* is currently the basis for most studies of archosauriform relationships (e.g., *Nesbitt & Butler, 2013*; *Butler et al., 2014*). This study utilizes the format used in that study for the listing of terminal taxa and characters to make this work compatible.

### *Adamanasuchus eisenhardtae* (*Lucas, Hunt & Spielmann, 2007*)

*Holotype* – PEFO 34638, partial skeleton including paramedian and lateral osteoderms, several vertebral centra, and a partial femur (*Lucas, Hunt & Spielmann, 2007*).

   *Referred Material* – PEFO 35093, osteoderm fragments, nasal fragment; PEFO 36806, osteoderm fragments.

   *Remarks* – *Lucas, Hunt & Spielmann (2007)* refer a lateral osteoderm (UCMP 126867) to *Adamanasuchus eisenhardtae* without explanation other than noting a 2007 personal communication from Andrew Heckert. They neither figure nor describe the specimen, but list its provenance as the *Placerias* Quarry near St. Johns, Arizona and attribute it as another Adamanian record of *Adamanasuchus eisenhardtae.* Examination of UCMP 126867 confirms the identification of the element as an aetosaurian lateral osteoderm; however, the specimen was collected from PFV 075 (Karen's Point) in Petrified Forest National Park and not from the *Placerias* Quarry. PFV 075 is in the Martha's Butte beds of the Sonsela Member, which are Revueltian in age (*Parker & Martz, 2011*), thus this would represent a range extension of this taxon up into the Sonsela Member and into the Revueltian biozone. This specimen differs from the holotype of *Adamanasuchus eisenhardtae* in possessing an extremely reduced dorsal flange and a dorsal eminence that forms a broadly triangular "spine" that projects dorsally. The outer surface of the lateral flange and the dorsal eminence bear an elongate ridge, which is located very close to the plate margin. Curiously the osteoderm lacks an anterior bar so it cannot be determined if this margin is the anterior or posterior edge. In *Adamanasuchus eisenhardtae*, the lateral osteoderms are more symmetrical with nearly equal lateral and dorsal flanges, and the eminence does not form a projected spine (PEFO 34638). Because of these anatomical differences and the discrepancy in the stratigraphic and locality data, the referral of this specimen to *Adamanasuchus eisenhardtae* is not supported.

   PEFO 35093 includes osteoderm fragments that possess the unique surface ornamentation of a faint background, radial pattern, 'overprinted' by deep randomly developed pits. This 'overprinting' is characteristic of *Adamanasuchus eisenhardtae* and differs from other aetosaurians with a radial ornamentation pattern

(*Lucas, Hunt & Spielmann, 2007*). An associated fragment of a nasal most likely belongs to the same specimen as it has an identical preservation and no other aetosaur specimens were recovered from the immediate area. Unfortunately, the nasal fragment is too incomplete to provide more information. PEFO 36806 is another referred specimen and consists solely of osteoderm fragments. Both PEFO 35093 and PEFO 36806 were recovered from the upper part of the Blue Mesa Member at about the same stratigraphic horizon as the holotype specimen of *Adamanasuchus eisenhardtae*.

*Age* – Late Triassic, early to middle Norian, Adamanian (*Ramezani et al., 2011*; *Parker & Martz, 2011*).

*Occurrence* – upper Blue Mesa Member, Chinle Formation, Petrified Forest National Park, Arizona, U.S.A. (*Lucas, Hunt & Spielmann, 2007*; *Parker & Martz, 2011*).

*Remarks* – *Lucas, Hunt & Spielmann (2007)* named *Adamanasuchus eisenhardtae* for a partial skeleton collected from the upper part of the Blue Mesa Member (Chinle Formation) in Petrified Forest National Park in 1996 (*Hunt, 1998*; *Parker, 2006*). *Parker (2006)* incorrectly assigned this specimen to *Typothorax antiquum* based on interpretation of comments made by *Hunt (1998)* regarding this specimen. In 2010, park staff revisited the type locality and finished the excavation; several paramedian and lateral osteoderms had been covered and left by the original workers and these materials were not included in the original description. The diagnosis provided by *Lucas, Hunt & Spielmann (2007)* does not adequately differentiate *Adamanasuchus eisenhardtae* from other known aetosaurians, in particular from *Calyptosuchus wellesi*; however, key characters found in *Adamanasuchus eisenhardtae* to the exclusion of *Calyptosuchus wellesi* are the strongly sigmoidal lateral edge, that results is a ventrolateral corner of the osteoderm that appears to have been sheared-off, and a triangular patch in the posteromedial corner of the paramedian osteoderm surface that is smooth and devoid of ornamentation. The first character state also occurs in paratypothoracins and the second is found in a new aetosaur species described below (e.g., PEFO 34616), except that in the latter taxon the triangular area is strongly raised.

*Key References* – *Lucas, Hunt & Spielmann (2007)*.

### *Aetobarbakinoides brasiliensis* (*Desojo, Ezcurra & Kischlat, 2012*)

*Holotype* – CPE2 168, partial postcranial skeleton (*Desojo, Ezcurra & Kischlat, 2012*). A cast of this specimen is in the Petrified Forest National Park (PEFO) collections.

*Referred Material* – none.

*Age* – Late Triassic, late Carnian – earliest Norian, *Hyperodapedon* Assemblage Zone (*Langer et al., 2007*; *Martinez et al., 2011*).

*Occurrence* – Sequence 2, Santa Maria Supersequence, Rio Grande Do Sul, Brazil (*Desojo, Ezcurra & Kischlat, 2012*).

*Remarks* – The holotype (CPE2 168) of *Aetobarbakinoides brasiliensis* is a fragmentary postcranial skeleton of a small aetosaurian that was originally referred to *Stagonolepis robertsoni* (=*Aetosauroides* in their hypothesis) by *Lucas & Heckert (2001)*. The lack of open neurocentral sutures in the cervical and trunk vertebrae suggests that CPE2 168

represents a skeletally mature individual (*Irmis, 2007*). Despite the fragmentary preservation of the holotype, *Desojo & Ezcurra (2011)* were able to distinguish this material from that of other South American aetosaurs, based on the presence of discrete vertebral laminae in the trunk series, a character lacking in taxa such as *Aetosauroides scagliai* and *Neoaetosauroides engaeus*. Furthermore, *Aetobarbakinoides* is the only South American aetosaurian specimen with trunk vertebrae that bear accessory articular structures (i.e. hyposphene), a feature recognized previously in aetosaurians only in desmatosuchines (*Parker, 2008b*). Determining the phylogenetic position of this taxon is difficult because it is represented primarily by endoskeletal (non-osteoderm) material. A few osteoderms are present, but the surface ornamentation is poorly preserved. Lateral osteoderms, which have been key to phylogenetic placement (*Parker, 2007*), are not preserved. Furthermore, the preserved paramedian osteoderms lack their lateral edges, which, if preserved, would have provided information about the medial edges of the lateral osteoderms allowing for the scoring of some characters. *Desojo, Ezcurra & Kischlat (2012)* recovered *Aetobarbakinoides brasiliensis* as the sister taxon of the clade Desmatosuchinae + Typothoracinae; however, *Heckert et al. (2015)* considered it to be a 'wildcard' (unstable) taxon in their analysis and pruned it *a posteriori* from their published tree. It performed as a wildcard taxon in the present analysis as well, which is discussed in more detail below.

Key References – *Desojo, Ezcurra & Kischlat (2012)*.

### *Aetosauroides scagliai* (*Casamiquela, 1960*)

*Holotype* – PVL 2073, postcranial skeleton including the majority of the carapace, vertebral column, and sacrum in articulation (*Casamiquela, 1961*).

*Referred Material* – see *Desojo & Ezcurra (2011)*.

*Age* – Late Triassic, Carnian, *Hyperodapedon* Assemblage Zone (*Rogers et al., 1993*; *Furin et al., 2006*; *Martinez et al., 2011*).

*Occurrence* – Cancha de Bochas Member, Ischigualasto Formation, Argentina; Sequence 2, Santa Maria Supersequence, Rio Grande do Sul State, Brazil (*Casamiquela, 1961*; *Desojo & Ezcurra, 2011*).

*Remarks* – *Aetosauroides scagliai* was originally described by *Casamiquela (1960)* & *Casamiquela (1961)* based on well-preserved cranial and postcranial material from the lower part of the Ischigualasto Formation of Argentina. Further material was assigned by *Casamiquela (1967)* who redescribed the specimens in light of the monograph on *Stagonolepis robertsoni* by *Walker (1961)*. Strong similarities have been noted between *Aetosauroides* and *Stagonolepis* as well as *Aetosaurus* and based on element size *Aetosauroides* was considered to be somewhat morphologically transitional between the two European taxa (*Casamiquela, 1967*). In an unpublished masters thesis, *Zacarias (1982)* erected a second species of *Aetosauroides* ("*Aetosauroides subsulcatus*") for material from the Upper Triassic of Brazil. All of this material has been briefly redescribed, the majority of it assigned to *Stagonolepis robertsoni* (*Lucas & Heckert, 2001*; *Heckert & Lucas, 2002*). Those authors argued that only superficial differences could be found between all of these specimens and that assignment of the South American material strengthened previously proposed biostratigraphic correlations between Brazil, Argentina, and the U.K.,

as well as to the southwestern United States. In contrast, *Desojo & Ezcurra (2011)* assigned the Brazilian material to *Aetosauroides scagliai* based on the presence of well-developed fossae on the lateral sides of the trunk vertebrae and the exclusion of the maxilla from the external naris in the skull of *Aetosauroides scagliai*, a character first noted by *Casamiquela (1967)*. A phylogenetic analysis recovered *Aetosauroides scagliai* as the sister taxon to all other aetosaurs (Stagonolepididae) (*Desojo, Ezcurra & Kischlat, 2012*). Redescriptions of the Argentinian material were presented in two unpublished dissertations (*Desojo, 2005*; *Parker, 2014*), and a full redescription by Desojo and Ezcurra is in progress (J. Desojo, personal communication, 2014).

The cranial material of *Aetosauroides scagliai* is significant because it exemplifies the plesiomorphic aetosaurian skull condition, optimizing characters such as the exclusion of the maxilla from the external naris, frontals that are wider than the parietals, nasals that taper anteriorly, a large triangular depression present anterior to the frontals, the lack of a 'slipper-shaped' mandible, the lack of a basal swelling in the teeth, and the mediolaterally compressed teeth with recurved tips (*Parker, 2014*). The skull is significantly different from that of *Stagonolepis robertsoni*, *Stagonolepis olenkae*, *Neoaetosauroides engaeus*, and *Calyptosuchus wellesi* and that characters of the osteoderms used to unite these taxa (e.g., *Heckert & Lucas, 2002*) are homoplasious (*Desojo & Ezcurra, 2011*; *Parker, 2008b*).

*Cerda & Desojo (2011)* provide details of the osteoderm histology of *Aetosauroides scagliai*, although using referred specimens rather than the holotype. This adds to the increasing understanding of the bone histology of aetosaurians (e.g., *Parker, Stocker & Irmis, 2008*; *Scheyer, Desojo & Cerda, 2013*). It is possible that once histological features and their relationships with ontogenetic maturity at time of death and potential environmental effects are better known, that histological characters can be incorporated in phylogenetic analyses of the Aetosauria.

*Key References* – *Casamiquela (1960)*; *Casamiquela (1961)*; *Casamiquela (1967)*; *Heckert & Lucas (2002)*; *Desojo & Ezcurra (2011)*; *Cerda & Desojo (2011)*; *Parker (2014)*.

### *Aetosaurus ferratus* (*Fraas, 1877*)

*Lectotype* – SMNS 5770, specimen XVI (16) (*Schoch, 2007*).

*Referred Material* – SMNS 5770, at least 24 specimens recovered in the same block as the lectotype; SMNS 18554, articulated skeleton lacking the skull and pectoral girdle; SMNS 14882, articulated caudal segment; SMNS 12670, trunk and ventral osteoderms; MCZ 22/92G, partial skull, limb bones and vertebrae, osteoderms; MCSNB 4864, trunk osteoderms.

*Age* – Late Triassic, middle Norian to early Rhaetian, Revueltian (*Deutsche Stratigraphische Kommission, 2005*; *Lucas, 2010*).

*Occurrence* – Lower and Middle Stubensandstein, Löwenstein Formation, Germany; Calcare de Zorzino Formation, Italy; Ørsted Dal Member, Fleming Fjord Formation, eastern Greenland (*Wild, 1989*; *Jenkins et al., 1994*; *Schoch, 2007*).

*Remarks* – The genus *Aetosaurus* originally included two species, *Aetosaurus ferratus* and *Aetosaurus crassicauda*. *Aetosaurus crassicauda* is presently understood to represent a larger specimen of *Aetosaurus ferratus* (*Schoch, 2007*). Specimens of *Stegomus arcuatus*

from eastern North American have been assigned to *Aetosaurus* (*Lucas, Heckert & Huber, 1998*); however, the majority of this material consists of natural molds that do not preserve the surface ornamentation. These specimens are assignable to *Aetosaurus* only on the basis of "aetosaurine" (*sensu Parker, 2007*) synapomorphies such as a sigmoidal lateral margin of the paramedian osteoderms with a pronounced anterolateral projection, as well as their small size. Small osteoderms (e.g., NMMNH P-17165) from the Bull Canyon Formation of New Mexico referred to *Stegomus* (*Aetosaurus*) *arcuatus* by *Heckert & Lucas (1998)* possess an anterior bar, radial pattern, offset dorsal eminence, and an anterolateral projection, which are "aetosaurine" characters and not diagnostic of a less inclusive taxon. Several authors consider the lack of dorsal ornamentation, including a dorsal eminence (boss) in the osteoderms of *Stegomus* (*Aetosaurus*) *arcuatus* to be diagnostic of the taxon (e.g., *Heckert & Lucas, 2000*; *Heckert et al., 2001*; *Spielmann & Lucas, 2012*); however, the lack of ornamentation is because the type and key referred specimens consist solely of natural molds of the ventral surfaces of the osteoderms which are typically smooth and unornamented in aetosaurs.

Purported specimens of *Aetosaurus ferratus* from the Chinle Formation of Colorado (*Small, 1998*) are now considered to represent a distinct taxon, *Stenomyti huangae* (*Small & Martz, 2013*). *Aetosaurus* has also been recognized from Greenland and Italy. The Greenland material consists of a partial skull, postcranial skeleton and osteoderms (MCZ 22/92G; *Jenkins et al., 1994*). This skull possesses the following characteristics of *Aetosaurus ferratus*; an anteroposteriorly short jugal, a round supratemporal fenestra; and an antorbital fossa that covers the majority of the lacrimal (*Schoch, 2007*). The Italian material (MCSNB 4864) consists of a short series of articulated dorsal paramedian and lateral osteoderms that possess an identical surface ornamentation to *Aetosaurus ferratus* (*Wild, 1989*). This specimen is significant as it was recovered from marine sediments of Norian age and represents a potential tie point to the marine biostratigraphic record for the Late Triassic (*Lucas, 1998a*; *Irmis et al., 2010*).

In summary, *Aetosaurus ferratus* is presently known from Greenland, Germany, and Italy, and other purported North American occurrences cannot be substantiated (*Schoch, 2007*; *Small & Martz, 2013*). For this study *Aetosaurus ferratus* is scored only from the German lectotype and referred material.

*Key References* – *Wild (1989)*; *Jenkins et al. (1994)*; *Schoch (2007)*.

### *Apachesuchus heckerti* (*Spielmann & Lucas, 2012*)

*Holotype* – NNMNH P-31100, left dorsal paramedian osteoderm.

*Referred material* – NMMNH P-63427, left cervical paramedian osteoderm; NMMNH P-63426, right caudal paramedian osteoderm. Both of these specimens were originally included in NMMNH P-31100 (*Heckert et al., 2001*; *Spielmann & Lucas, 2012*:fig. 70e), but have been renumbered. *Spielmann & Lucas (2012)* also report that much more complete material of this taxon, including postcrania, is currently under study by Axel Hungerbühler at the Mesalands Dinosaur Museum in Tucumcari, New Mexico. This new material is also from the Redonda Formation of New Mexico; however, the new material referable to *Apachesuchus heckerti* only consists of a few more paramedian osteoderms,

whereas the rest of the material is actually referable to *Redondasuchus rineharti* (J. Martz, personal communication, 2013).

*Age* – Late Triassic, late Norian-Rhaetian, Apachean (*Spielmann & Lucas, 2012*).

*Occurrence* – Quay Member, Redonda Formation, Dockum Group, New Mexico, U.S.A (*Spielmann & Lucas, 2012*).

*Remarks* – The holotype and paratype (referred) osteoderms were recovered in a microvertebrate assemblage found within a very large phytosaur skull and were originally assigned to *Neoaetosauroides* sp. because the lack of surface ornamentation of the paramedian osteoderms was thought to be diagnostic of *Neoaetosauroides* (*Heckert et al., 2001*). However, the lack of surface ornamentation of some of the osteoderms of the holotype of *Neoaetosauroides* is the result of overpreparation of the specimen and close examination shows that the material does have a surface orientation of radial grooves and ridges; therefore the NMMNH material cannot be assigned to that taxon. The lack of surface ornamentation in the type material of *Apachesuchus heckerti* appears to be a genuine feature and is considered an autapomorphy of the taxon (*Spielmann & Lucas, 2012*; J. Martz, personal communication, 2013). *Apachesuchus heckerti* is considered to possess a low width/length ratios (> 0.3) of the paramedian osteoderms; which was obtained by comparing the length of the lateral edge to the total plate length (*Heckert et al., 2001*; *Spielmann & Lucas, 2012*). However, the lateral edge of NMMNH P-31100 is greatly expanded anteroposteriorly than the rest of the osteoderm strongly skewing this ratio. The length at the center of the osteoderm is 32 mm, compared to an overall width of 104 mm. This provides a width/length ratio of 3.25, compared to the ratio of 2.5 provided by *Spielmann & Lucas (2012)*. It is important to standardize areas of measurements when determining ratios of aetosaur osteoderms because simply using maximum length can skew results in osteoderms with abnormal shapes. This is also true for osteoderms with elongate anterolateral processes of the anterior bars (e.g., *Calyptosuchus wellesi*). In these cases osteoderm lengths should be taken from the main osteoderm body and not from the anterior bar. Furthermore, an unnumbered referred anterior dorsal paramedian osteoderm in the Mesalands Community College Dinosaur Museum (MCCDM) collection (field number 20080618RET002RRB) has a width of 110 mm and a median length of 28 mm for a W/L ratio of 3.92. This compares well with typothoracine aetosaurs such as *Typothorax coccinarum* (*Long & Murry, 1995*; *Heckert et al., 2010*).

*Key References* – *Heckert et al. (2001)*; *Spielmann & Lucas (2012)*.

### *Calyptosuchus wellesi* (*Long & Ballew, 1985*)

*Holotype* – UMMP 13950, articulated carapace from the posterior dorsal and caudal regions, associated with a portion of the vertebral column and the sacrum (*Case, 1932*; *Long & Ballew, 1985*).

*Referred Material* – UMMP 7470, two trunk paramedian osteoderms, three trunk vertebrae, mostly complete, articulated sacrum; UCMP 27225, paramedian, lateral, and ventral osteoderms, partial right dentary. Numerous specimens from the *Placerias* Quarry from the UCMP and the MNA collections, as well as specimens from Petrified Forest

National Park also can be referred to *Calyptosuchus wellesi* (*Long & Murry, 1995*; *Parker, 2014*).

*Age* – Late Triassic, early-middle Norian, early Adamanian (*Ramezani et al., 2011*; *Ramezani, Fastovsky & Bowring, 2014*; *Parker & Martz, 2011*).

*Occurrence* – upper Blue Mesa Member, Chinle Formation, Arizona, U.S.A.; Tecovas Formation, Dockum Group, Texas, U.S.A (*Long & Murry, 1995*; *Parker & Martz, 2011*).

*Remarks* – *Case (1932)* described a posterior portion of a carapace and associated pelvis and vertebral column of what he believed to be a phytosaur from the Upper Triassic of Texas. Although he discussed possible taxonomic affinities he was thoroughly perplexed by the material and thus did not assign the specimen to an existing taxon or coin a new taxonomic name. This is mainly because of the common association of aetosaurian osteoderms with phytosaur remains (e.g., *Nicrosaurus kapffi*, *Case, 1929*) and because the osteoderms of UMMP 13950 possessed a radial surface ornamentation more similar to osteoderm material then assigned to the phytosaur *Nicrosaurus* (=*Phytosaurus*) *kapffi* (now the holotype of the aetosaurian *Paratypothorax andressorum Long & Ballew, 1985*). This is unlike the surface ornamentation found in the other aetosaurian Case was familiar with, *Desmatosuchus spurensis* (*Case, 1922*). Indeed, *Case (1932)* tentatively suggested that UMMP 13950 may belong to the genus "*Phytosaurus*." *Gregory (1953a)* recognized that the specimen was probably more closely related to *Typothorax* than to phytosaurs and hence most likely a pseudosuchian (aetosaur), but still considered the purported close similarity of the rectangular osteoderms with those assigned to some phytosaurs to be problematic for taxonomic resolution of the material.

This problem was finally resolved by *Long & Ballew (1985)* who correctly determined that all of the Triassic material with broad, rectangular osteoderms was referable to aetosaurians. Those authors also listed UMMP 13950 as the holotype of a new genus, *Calyptosuchus wellesi*. They did not redescribe Case's specimen, but instead discussed the new taxon in terms of referred material from the Triassic of Arizona. A recent description of the taxon is by *Long & Murry (1995)* who mainly described referred material from the *Placerias* Quarry of Arizona. The referrals of material to *Calyptosuchus wellesi* by *Long & Murry (1995)* have been questioned mainly because of the recognition that the cervical lateral osteoderms assigned to *Calyptosuchus wellesi* by *Long & Ballew (1985)* & *Long & Murry (1995)* actually belong to a paratypothoracin aetosaur demonstrating the presence of a third aetosaur taxon in the *Placerias* Quarry (*Parker, 2005a*; *Parker, 2007*).

*Parker (2014)* carefully sorted and grouped the *Placerias* Quarry material based on field numbers and used the resulting associations as well as apomorphic comparisons to test these assignments. Referred elements of *Calyptosuchus wellesi* were redescribed and these referred specimens, as well as the holotype, are used to and score that taxon in this phylogenetic analysis. This anatomical work, in association with detailed biostratigraphic work of the Chinle Formation (*Parker & Martz, 2011*), has also determined that *Calyptosuchus wellesi* is presently restricted to the upper part of the Blue Mesa Member and that specimens of *Calyptosuchus* noted from the Sonsela Member (e.g., *Parker & Martz, 2011*) belong to a new taxon described below.
Key References – *Case (1932)*; *Long & Ballew (1985)*; *Long & Murry (1995)*; *Parker (2014)*.

### Coahomasuchus kahleorum (*Heckert & Lucas, 1999*)

*Holotype* – NMMNH P-18496, much of an articulated, but crushed skeleton (*Heckert & Lucas, 1999*).

*Referred Material* – TMM 31100-437, partial skull, paramedian, lateral, and ventral osteoderms, vertebrae, limb, and girdle material (*Murry & Long, 1996*; this study); NCSM 23168, much of a carapace (*Heckert et al., 2015*).

*Age* – Late Triassic, ?Carnian, Otischalkian (*Lucas, 2010*).

*Occurrence* – Colorado City Formation, Dockum Group, west Texas, U.S.A.; Pekin Formation, Newark Supergroup, North Carolina, U.S.A (*Heckert & Lucas, 1999*; *Heckert et al., 2015*).

*Remarks* – The holotype of *Coahomasuchus kahleorum* is distinctive, but poorly preserved, consisting of a flattened carapace and plastron concealing the majority of the vertebrae, the posteroventral corner of the skull, the posterior portion of the mandible, and a poorly preserved braincase, as well as articulated limb and girdle material (*Heckert & Lucas, 1999*; *Desojo & Heckert, 2004*). *Fraser et al. (2006)* documented the first occurrence of *Coahomasuchus* in the Pekin Formation of North Carolina providing a biostratigraphic correlation with the lower part of the Dockum Group of west Texas. Past phylogenetic analyses have recovered *Coahomasuchus kahleorum* as the sister taxon of *Typothorax coccinarum* and *Redondasuchus reseri* (*Harris, Gower & Wilkinson, 2003* correction of the *Heckert & Lucas, 1999* dataset); as the sister taxon of an unresolved clade containing *Aetosauroides*, *Calyptosuchus*, *Aetosauroides*, and *Aetosaurus* (*Parker, 2007*); and in an unresolved position closer to the base of Stagonolepididae (*Desojo, Ezcurra & Kischlat, 2012*). Moreover, the latter authors pruned *Coahomasuchus* from their final tree to achieve better resolution, thus the phylogenetic relationships of this taxon are far from resolved. However, a more recent analysis by *Heckert et al. (2015)*, utilizing a modified version of the dataset in *Parker (2007)* & *Desojo, Ezcurra & Kischlat (2012)*, recovered *Coahomasuchus* as a non-stagonolepidid aetosaur at the base of Aetosauria. In this analysis *Coahomasuchus kahleorum* is coded from the holotype as well as a newly referred specimen from the Dockum Group of Texas (TMM 31100-437) previously referred to as the 'carnivorous form' (*Murry & Long, 1996*), which was recovered from the same geographical area and stratum as the type specimen (*Lucas, Hunt & Kahle, 1993*).

It was suggested that the holotype of *Coahomasuchus kahleorum* may represent a skeletally immature individual (*Parker, 2003*). However, histological sampling of the referred specimen TMM 31100-437, which is in the same size class, indicates that TMM 31100-437 is close to skeletal maturity (S. Werning, personal communication, 2014). Thus, *Coahomasuchus kahleorum* is most likely not a juvenile individual of *Lucasuchus hunti* or *Longosuchus meadei*, both which are found in the same stratigraphic horizon and localities (e.g., *Parker & Martz, 2010*).

Key References – *Heckert & Lucas (1999)*; *Desojo & Heckert (2004)*.

### *Desmatosuchus spurensis* (*Case, 1920*)

*Holotype* – UMMP 7476, skull, nearly complete carapace, articulated cervical and dorsal vertebral column, ilium (*Case, 1922*).

*Referred Material* – see *Parker (2008b)*.

*Age* – Late Triassic, early to middle Norian, Adamanian (*Ramezani et al., 2011*; *Ramezani, Fastovsky & Bowring, 2014*; *Parker & Martz, 2011*).

*Occurrence* – Tecovas Formation, Dockum Group, Texas, U.S.A., Los Esteros Member, Santa Rosa Formation, Dockum Group, New Mexico, U.S.A., upper Blue Mesa Member, Chinle Formation, Arizona, U.S.A (*Long & Murry, 1995*; *Parker, 2008b*).

*Remarks* – First described from much of a carapace, and associated vertebral column as well as a skull, *Desmatosuchus spurensis* is a well-known aetosaurian from the Upper Triassic of the southwestern United States. Despite this, confusion exists regarding characters of the dorsal armor for referral of specimens. For example all of the specimens listed by *Long & Ballew (1985)* from Petrified Forest National Park actually pertain to paratypothoracines, and the osteoderm of *Desmatosuchus* figured by *Lucas & Connealy (2008*:26*)* for the Dawn of the Dinosaurs exhibit at the New Mexico Museum of Natural History and Science is actually and osteoderm of *Calyptosuchus wellesi*.

*Gregory (1953a)* synonymized *Desmatosuchus spurensis* with *Episcoposaurus haplocerus*, a form described by *Cope (1892)*, and the taxon was known as *Desmatosuchus haplocerus* for several decades, until it was determined that *Episcoposaurus haplocerus* was actually a *nomen dubium* (*Parker, 2008b*; *Parker, 2013*) although this has not been accepted by all workers (e.g., *Heckert, Lucas & Spielmann, 2012*). New material from the Chinle Formation of Arizona demonstrated that previous carapace reconstructions for *Desmatosuchus spurensis* were erroneous and the body was broader than previous believed (*Parker, 2008b*).

Limb and pectoral girdle for *Desmatosuchus spurensis* is not known from the two best preserved specimens (UMMP 7476, MNA V9300), but *Long & Murry (1995)* assigned isolated material from the *Placerias* Quarry to the taxon, which has been utilized for studies including bone histology (*de Ricqlès, Padian & Horner, 2003*). Unfortunately *Long & Murry (1995)* did not discuss the evidence for these referrals, which have been questioned (*Parker, 2005a*; *Parker, 2008b*); however, utilizing field numbers from the *Placerias* Quarry it may possible to refer some of this material to *Desmatosuchus spurensis*. For this analysis *Desmatosuchus spurensis* is coded from UMMP 7476 and MNA V9300.

*Key References* – *Case (1920)*; *Case (1922)*; *Long & Ballew (1985)*; *Long & Murry (1995)*; *Parker (2008b)*.

### *Desmatosuchus smalli* (*Parker, 2005b*)

*Holotype* – TTU P-9024, almost complete skull and right mandible, partial pelvis, femora, nearly complete cervical armor and numerous osteoderms from the rest of the carapace (*Parker, 2005b*).

*Referred Material* – see *Parker (2005b)* & *Martz et al. (2013)*.

*Age* – Late Triassic, mid-Norian, latest Adamanian and possibly earliest Revueltian (*Ramezani et al., 2011*; *Martz et al., 2013*).

*Occurrence* – Middle section of the Cooper Canyon Formation, Dockum Group, Texas, U.S.A.; ?Martha's Butte beds, Sonsela Member, Chinle Formation, Arizona, U.S.A (*Parker, 2005b*; *Martz et al., 2013*).

*Remarks* – *Small (1985)* & *Small (2002)* described new material of *Desmatosuchus* from the Cooper Canyon Formation of Texas. Although he noted differences in the cranial material of the new material from the holotype of *Desmatosuchus spurensis* (UMMP 7476), he did not feel they were of taxonomic significance. In a revision of the genus *Desmatosuchus*, significant differences in the lateral armor were noted between the Cooper Canyon specimens and the type of *Desmatosuchus spurensis* (*Parker, 2003*). Combined with the cranial differences noted by *Small (2002)* the Cooper Canyon Formation material was assigned to a new species (*Parker, 2005b*). Further comments regarding this taxon including a novel reconstruction of the lateral cervical armor were provided by *Martz et al. (2013)*.

One of the problems in utilizing the non-osteoderm postcranial material of *Desmatosuchus smalli* is that some of it may actually pertain to an undescribed specimen of *Paratypothorax* from the quarry (*Martz, 2008*). A detailed apomorphy-based study of the aetosaurian material from the Post Quarry is needed along with a reinvestigation of field collection data to clarify some of the taxonomic assignments of the material (*Martz, 2008*).

Other than the Texas material, *Desmatosuchus smalli* is known from only one single referred lateral osteoderm from the Chinle Formation of Arizona (MNA V697), which had been assigned to *Desmatosuchus* by *Long & Ballew (1985)* as a cervical lateral osteoderm. MNA V697 actually represents a dorsal lateral osteoderm and is assigned to *Desmatosuchus smalli* based on the ventrally recurved spine tip, which is an autapomorphy of *Desmatosuchus smalli* and does not occur in *Desmatosuchus spurensis* (*Parker, 2005b*). Although MNA V697 is listed as originating from a locality in the upper part of the Sonsela Member near Petrified Forest National Park (*Long & Ballew, 1985*), the locality data for this specimen are ambiguous. However, if correct this would represent the only Revueltian occurrence of *Desmatosuchus* (*Parker & Martz, 2011*).

The holotype of *Desmatosuchus* (=*Episcoposaurus*) *haplocerus* (ANSP 14688; *Cope, 1892*) consists chiefly of lateral and paramedian osteoderms of the cervical and anterior trunk regions (*Gregory, 1953a*; *Parker, 2013*). Unfortunately the tips of the spines on all of the trunk lateral osteoderms are broken away so the material cannot be differentiated between *Desmatosuchus spurensis* and *Desmatosuchus smalli*. Interestingly, the shape of the cervical lateral osteoderms as well as the ornamentation of the trunk paramedian osteoderms are more reminiscent of *Desmatosuchus smalli* rather than *Desmatosuchus spurensis*, but the data are not conclusive and therefore *Desmatosuchus haplocerus* is considered a *nomen dubium* (*Parker, 2008b*; *Parker, 2013*).

*Key References* – *Small (1985)*; *Small (2002)*; *Parker (2005b)*; *Martz et al. (2013)*.

### Longosuchus meadei (*Sawin, 1947*)

*Lectotype* – TMM 31185-84b, skull and much of a postcranial skeleton (*Sawin, 1947*). See *Parker & Martz (2010)* for detailed discussion of the status of the type materials.

*Referred Material* – TMM 31185-84a, partial skull and postcranial skeleton. See *Long & Murry (1995)* for a complete list.

*Age* – Late Triassic, ?Carnian, Otischalkian (*Lucas, 2010*).

*Occurrence* – Colorado City Formation, Dockum Group, Texas, U.S.A (*Hunt & Lucas, 1990*).

*Remarks* – The Works Progress Administration program in the 1930s made vast collections of vertebrate fossils from a series of quarries in strata of the Dockum Group in Howard County, Texas. This included several skeletons of an aetosaurian that *Sawin (1947)* described as a new species of *Typothorax, Typothorax meadei*. Several subsequent authors recognized the distinctiveness of this material (*Long & Ballew, 1985*; *Small, 1989*; *Murry & Long, 1989*) and the species was placed in a new genus, *Longosuchus*, by *Hunt & Lucas (1990)*. Sawin's original description is thorough, but affected by a lack of good comparative material as well as the poor historical understanding of the taxonomic make-up of the Aetosauria available at the time of his initial work. Thus he incorrectly reconstructed the incomplete lower jaw and pelvis, which confused aetosaur in-group relationships until these details were later corrected by *Walker (1961)*.

Most of the Otis Chalk material remains unprepared and numerous specimens, including partial skeletons (unpublished TMM documents), referable to *Longosuchus meadei* are in the Vertebrate Paleontology Lab (VPL) collections at the University of Texas (Austin) awaiting preparation.

An isolated fragment of a paramedian osteoderm from the Salitral Shale (Chinle Formation) of New Mexico, assigned to *Longosuchus meadei* by *Hunt & Lucas (1990)*, possesses a beveled posterior edge and a radial ornament pattern and is more likely referable to Paratypothoracini, in particular *Tecovasuchus* (*Irmis, 2008*). Lateral osteoderms from the Argana Group of Morocco assigned to *Longosuchus meadei* by *Lucas (1998b)* appear to also represent a paratypothoracin as they are strongly dorsoventrally compressed and slightly recurved (*Parker & Martz, 2010*). Unfortunately this cannot be tested as these specimens have been reported as lost (S. Nesbitt, personal communication, 2013). Character state scorings for this study for *Longosuchus* were made solely utilizing the TMM material.

*Key References* – *Sawin (1947)*; *Hunt & Lucas (1990)*; *Long & Murry (1995)*; *Parker & Martz (2010)*.

### *Lucasuchus hunti* (*Long & Murry, 1995*)

*Holotype* – TMM 31100-257, posterior trunk paramedian osteoderm (*Long & Murry, 1995*).

*Referred Material* – see *Parker & Martz (2010)* & *Long & Murry (1995)*.

*Age* – Late Triassic, ?Carnian, Otischalkian (*Lucas, 2010*).

*Occurrence* – Colorado City Formation, Dockum Group, Texas, U.S.A.; Pekin Formation, Newark Supergroup, North Carolina, U.S.A (*Long & Murry, 1995*; *Parker & Martz, 2010*).

*Remarks* – *Long & Murry (1995)* recognized the presence of two distinct large aetosaurian morphotypes in material from the Otis Chalk quarries in Howard County,

Texas, the first being *Longosuchus meadei* and a second for which they coined a new taxon, *Lucasuchus hunti*. *Sawin (1947)* had also recognized the presence of this second aetosaurian, which he erroneously assigned to *Typothorax coccinarum*. *Hunt & Lucas (1990)* overlooked *Sawin's (1947)* separation of the material when they reassigned all of the material to *Longosuchus meadei*. Separated out again by *Long & Murry (1995)*, the presence of two distinct taxa was disputed by some workers (e.g., *Heckert & Lucas, 1999*; *Heckert & Lucas, 2000*) until *Parker & Martz (2010)* presented the differences in greater detail (*Heckert et al., 2015*).

The holotype of *Lucasuchus hunti* is a single paramedian plate, but *Long & Murry (1995)* assigned numerous postcranial elements to the taxon. However, lack of preparation of much of this material, questions regarding associated with apomorphic osteoderms, as well as apparent similarities with *Longosuchus meadei* makes many of these referrals questionable. Nonetheless there is still much unprepared material at the VPL that is almost certainly represents *Lucasuchus hunti*. A recently prepared partial skull (TMM 31100-531) from Howard County, Texas differs in some ways from the lectotype skull of *Longosuchus meadei* and could represent *Lucasuchus hunti* (J. Martz, personal communication, 2008).

Osteoderms previously referred to *Desmatosuchus* and *Longosuchus* from the Pekin Formation of North Carolina actually pertain to *Lucasuchus* providing an important biostratigraphic correlation (*Parker & Martz, 2010*; *Heckert et al., 2015*).

*Key References – Long & Murry (1995)*; *Parker & Martz (2010)*; *Heckert et al. (2015)*.

### *Gorgetosuchus pekinensis* (*Heckert et al., 2015*)

*Holotype* – NCSM 21723, a large portion of the cervical and anterior trunk carapace.

*Referred Material* – none.

*Age* – Late Triassic, ?Carnian, Otischalkian (*Huber, Lucas & Hunt, 1993*).

*Occurrence* – Upper portion of the Pekin Formation, Newark Supergroup, North Carolina, U.S.A. (*Heckert et al., 2015*).

*Remarks* – The holotype of *Gorgetosuchus pekinensis* (NCSM 21723) consists solely of the anterior portion of the trunk carapace of a desmatosuchine aetosaur. Similar in overall anatomy to *Longosuchus meadei* and *Lucasuchus hunti*, *Gorgetosuchus pekinensis* differs from these two taxa, and all other desmatosuchines, mainly in the possession of cervical paramedian osteoderms that are wider than long.

*Key References – Schneider, Heckert & Fraser (2011)*; *Heckert et al. (2015)*.

### *Neoaetosauroides engaeus* (*Bonaparte, 1969*)

*Holotype* – PVL 3525, lower jaw and postcranial skeleton (*Bonaparte, 1969*).

*Referred Material* – see *Desojo & Báez (2007)*.

*Age* – Late Triassic, middle Norian, early Revueltian (*Martinez et al., 2013*; *Kent et al., 2014*).

*Occurrence* – Upper part of the Los Colorados Formation, Argentina (*Desojo & Báez, 2005*). *Lucas (1998a)* considered the Los Colorados Formation equivalent to his Apachean 'Land Vertebrate Faunachron' and therefore Rhaetian, or at least latest Norian,

based on the presence of sauropodomorph dinosaurs and crocodyliform pseudosuchians. However, recent reexamination of strata in the Ischigualasto Basin, including a detailed paleomagnetic study, suggests instead that the vertebrate bearing portion of the Los Colorados may in fact be equivalent to the upper portion of the Sonsela Member of the Chinle Formation and thus Revueltian in age (*Martinez et al., 2013*; *Kent et al., 2014*).

*Remarks* – The holotype of *Neoaetosauroides engaeus* was diagnosed by *Bonaparte (1969)* and first described in detail by *Bonaparte (1972)*. Poorly understood for the purpose of prior phylogenetic analyses, the holotype and several referred skulls were recently redescribed by *Desojo & Báez (2005)* & *Desojo & Báez (2007)*. *Heckert & Lucas (2000)* considered the paramedian osteoderms to be almost completely devoid of ornamentation and this lack of ornamentation to be an autapomorphy of the taxon. However, personal examination of the type specimens shows that *Neoaetosauroides engaeus* possesses a clear radial ornamentation of the dorsal osteoderms (also see *Desojo & Báez, 2005*). Indeed, the ornamentation is indistinguishable from that of the Ischigualasto taxon *Aetosauroides scagliai*. Portions of the holotype carapace are devoid of ornamentation, but this is clearly the result of overpreparation of the material. Nonetheless, three small osteoderms from the Redonda Formation (Dockum Group) of New Mexico were assigned to *Neoaetosauroides* based upon a lack of distinct ornamentation (*Heckert et al., 2001*). These osteoderms subsequently became the holotype of a new taxon *Apachesuchus heckerti* (*Spielmann & Lucas, 2012*). Character state scorings for *Neoaetosauroides engaeus* are from the type and referred materials.

*Key References* – *Bonaparte (1969)*; *Bonaparte (1972)*; *Desojo & Báez (2005)*; *Desojo & Báez (2007)*.

### *Paratypothorax andressorum* (*Long & Ballew, 1985*)

*Holotype* – SMNS unnumbered, left trunk paramedian osteoderm (labeled L18 on red sticker) (*Long & Ballew, 1985*).

*Paratypes* – SMNS unnumbered, partial disarticulated carapace that includes the holotype osteoderm.

*Referred Material* – NHMUK R38070, posterior trunk vertebra (*Meyer, 1865*:pl. XXVII, figs. 1–3); NHMUK R38083, left trunk paramedian osteoderm; NHMUK R38085, partial right trunk paramedian osteoderm (*Meyer, 1865*:pl. XXVIII, figs. 4–6); NHMUK R38086, partial right paramedian osteoderm; NHMUK R38087, pathologic left mid-caudal paramedian osteoderm (*Meyer, 1865*:pl. XXVIII, figs. 7–9; NHMUK R38090, right trunk paramedian osteoderm, partial left trunk paramedian osteoderm, three partial right paramedian osteoderms, partial left lateral osteoderm, left lateral osteoderm, two partial paramedian osteoderms; SMNS 3285, partial paramedian osteoderm; SMNS 2958, three pathologic paramedian osteoderms (*Lucas, 2000*); SMNS 4345 left trunk lateral osteoderm; SMNS 4386, right trunk lateral osteoderm (*Meyer, 1861*:pl. XLIII, fig. 1).; SMNS 5721 right paramedian osteoderm (*Meyer, 1865*:Pl. XXVIII, figs. 1–3); YPM 3694, right trunk lateral osteoderm (*Gregory, 1953b*).

*Age* – Late Triassic, Norian, Revueltian (*Deutsche Stratigraphische Kommission, 2005*; *Lucas, 2010*).

*Occurrence* – Lower Stubensandstein, Löwenstein Formation, Baden-Württemberg, Germany (*Long & Ballew, 1985*).

*Remarks* – The SMNS collections possess numerous osteoderms including much of what appears to be a carapace of a single individual that have had a confusing taxonomic history. The osteoderms were collected with and considered to belong to the phytosaur *Nicrosaurus* (=*Belodon* = *Phytosaurus*) until the mid-1980s (*Long & Ballew, 1985*). This belief caused significant confusion regarding the taxonomy of phytosaur and aetosaur material (*Case, 1932*; *Gregory, 1962*; *Gregory & Westphal, 1969*). The issue was finally sorted out when *Long & Ballew (1985)* recognized that all of the broad rectangular osteoderms belonged to aetosaurs and coined the name *Paratypothorax addressorum* for the German osteoderms originally assigned to *Nicrosaurus*. The species epithet was correctly amended to *Paratypothorax andressorum* by *Lucas & Heckert (1996)* as the species was originally named in honor of the Andress family (Chris Andress was Chief Ranger at Petrified Forest National Park in 1985). *Long & Ballew (1985)* also noted material from southwestern North America that is referable to *Paratypothorax* although they were unsure that it represented the same species as the European material. This has led to two views regarding the assignment of the North American material; 1) that it is referable to *Paratypothorax andressorum* (*Hunt & Lucas, 1992*; *Heckert & Lucas, 2000*; *Lucas, Heckert & Rinehart, 2006*), or that it may represent a new taxon (*Long & Ballew, 1985*; *Long & Murry, 1995*). This is not yet resolved so they are treated here as two distinct taxa.

The German material has never actually been fully described and the present concept of *Paratypothorax* (*sensu Long & Murry, 1995*) is actually based on the referred North American material. There is also some confusion regarding the type specimens of *Paratypothorax andressorum*, with some workers treating a well-preserved carapace (SMNS unnumbered) as the holotype or as a syntype series for the taxon (e.g., *Hunt & Lucas, 1992*, *Lucas, Heckert & Rinehart, 2006*). However, *Long & Ballew (1985*:57*)* clearly identify a single osteoderm as the holotype so the other osteoderms in this specimen can be no more than paratypes (*Heckert & Lucas, 2000*).

An impression of a partial trunk paramedian osteoderm (MCZ field No. 23/92G) from Greenland was assigned to *Paratypothorax andressorum* (*Jenkins et al., 1994*). Although the specimen clearly possesses a raised anterior bar, radial pattern of pits and grooves, a dorsal eminence that contacts the posterior osteoderm margin, characteristic for paratypothoracins, the beveled posterior edge delineated by a distinct ridge is not a clear autapomorphy of *Paratypothorax andressorum* and thus this specimen should instead be assigned to Paratypothoracini (*Martz & Small, 2006*; *Desojo et al., 2013*). I have not examined the other three osteoderms mentioned by *Jenkins et al. (1994)* and assigned to *Paratypothorax andressorum*.

*Key References* – *Long & Ballew (1985)*.

### *Paratypothorax* sp.

*Referred Material* – PEFO 3004, associated osteoderms and vertebrae from the posterior trunk and anterior caudal regions (*Long & Murry, 1995*); FMNH PR1610, partial paramedian osteoderm (same specimen as PEFO 3004); DMNH 9942; partial postcranial

skeleton (*Long & Murry, 1995*); VRPH2, numerous paramedian and lateral osteoderms; see *Martz et al. (2013)* for additional specimens.

*Age* – Late Triassic, Adamanian-Revueltian, mid-Norian (*Ramezani et al., 2011*; *Parker & Martz, 2011*).

*Occurrence* – Chinle Formation, Arizona and New Mexico, U.S.A.; Dockum Group, Texas, U.S.A (*Long & Murry, 1995*; *Parker & Martz, 2011*; *Martz et al., 2013*).

*Remarks* – the presence of *Paratypothorax* material in North America was first recognized by *Long & Ballew (1985)* although they were unsure of its exact relationship with the German material, which they named *Paratypothorax andressorum*. Since that time numerous specimens referable to *Paratypothorax* sp. or Paratypothoracini have been collected from the Upper Triassic Chinle Formation and Dockum Group (see *Long & Murry, 1995*; *Parker & Martz, 2011*; *Martz et al., 2013* for lists). This material includes lateral osteoderms from the *Placerias* Quarry of Arizona that were originally identified by *Long & Ballew (1985)* as cervical laterals of *Calyptosuchus wellesi* (*Parker, 2005a*). The best preserved specimen of *Paratypothorax* sp. (PEFO 3004) is an associated set of posterior trunk and anterior caudal osteoderms and vertebrae of a single individual from the Chinle Formation of Arizona. First mentioned by *Long & Ballew (1985)*, but described by *Hunt & Lucas (1992)*, the latter authors assigned PEFO 3004 to *Paratypothorax andressorum*. This assignment was followed by *Heckert & Lucas (2000)* & *Lucas, Heckert & Rinehart (2006)*. However, differences between the North American and European material were noted by *Long & Murry (1995)* based on a specimen from the Dockum Group of Texas (DMNH 9942). Therefore the North American material is treated separately for this study. *Paratypothorax* sp. is known almost solely from osteoderms and vertebrae (*Hunt & Lucas, 1992*; *Long & Murry, 1995*). However, DMNH 9942 contains some forelimb material (*Long & Murry, 1995*). *Long & Murry (1995)* also questionably referred an ilium from the Post Quarry of Texas to the taxon, but this assignment is ambiguous. *Martz et al. (2013)* figure a fibula (TTU P-09416) they assign to *Paratypothorax* sp. A dentary of *Paratypothorax* was mentioned by *Small (1989)*; however, the specimen is now considered to be a lateral osteoderm (*Martz et al., 2013*). It is possible that cranial material referred by *Small (2002)* to *Desmatosuchus* actually represents *Paratypothorax* sp. (*Martz et al., 2013*), but this has not yet been fully demonstrated.

*Key References* – *Hunt & Lucas (1992)*; *Small (1989)*; *Long & Ballew (1985)*; *Long & Murry (1995)*; *Martz et al. (2013)*.

### *Polesinesuchus aurelioi (Roberto-Da-Silva et al., 2014)*

*Holotype*—ULBRA PVT003, parietal and braincase fragments, much of a postcranial skeleton (*Roberto-da-Silva et al., 2014*).

*Age* – Late Triassic, late Carnian – earliest Norian, *Hyperodapedon* Assemblage Zone (*Langer et al., 2007*; *Martinez et al., 2011*).

*Occurrence* – Sequence 2, Santa Maria supersequence, Rio Grande Do Sul, Brazil (*Desojo, Ezcurra & Kischlat, 2012*).

*Remarks*—*Polesinesuchus aurelioi* was erected for mainly the endoskeletal material of a skeletally immature aetosaurian from the Upper Triassic of Brazil

(*Roberto-da-Silva et al., 2014*). The taxon was not diagnosed by any recognized autapomorphies, but rather from a unique combination of characters that differentiates it from all known South American aetosaurians. Overall the material is most similar to that of *Aetosauroides scagliai*, but lacks the deep lateral fossae found in the cervical and trunk vertebrae of that taxon. The vertebrae of *Polesinesaurus aurelioi* are notable in that they appear to lack vertebral laminae, which may be an autapomorphy of the taxon. However, the laterally expansive prezygapophyses listed as a defining character of the taxon may actually represent prezygadiapophyseal laminae (*sensu Wilson, 1999*), as these laminae form a similar structure in the presacral vertebrae of *Scutarx deltatylus* (PEFO 31217). The skeletally immature status of the material is problematic because our present understanding of character variation and transformation through ontogeny is poor and these unique characteristics may simply be the result of the ontogenetic immaturity at time of death. Indeed, *Polosinesuchus aurelioi* appears to represent the well-preserved, but relatively unremarkable remains of a skeletal immature aetosaurian. Future histological studies of this taxon and others across will provide needed information on the timing of the appearance of key osteological landmarks in aetosaurian clades.

A recent phylogenetic analysis recovered *Polesinesuchus* as the sister taxon to *Aetobarbakinoides* in a clade that is sister taxon to Desmatosuchinae plus Typothoracinae, but this could be an artifact of missing data, especially from the paramedian and lateral osteoderms (*Roberto-da-Silva et al., 2014*).

*Key Reference* – *Roberto-da-Silva et al. (2014)*.

### *Postosuchus kirkpatricki* (*Chatterjee, 1985*)

*Holotype* – TTU P-9000, almost complete skull and partial skeleton (*Chatterjee, 1985*).

*Paratype* – TTU P-9002, almost complete skull and partial skeleton (*Chatterjee, 1985*).

*Age* – Late Triassic, early to middle Norian, Adamanian (*Martz et al., 2013*).

*Occurrence* – Cooper Canyon Formation, Dockum Group, Texas, U.S.A (*Martz et al., 2013*).

*Remarks* – *Postosuchus kirkpatricki* is a well-known rauisuchid archosaur represented by excellent material from the Post Quarry of Texas. The type materials were recently redescribed in detail by *Weinbaum (2011)* & *Weinbaum (2013)*. Technically because the species was named in honor of the Kirkpatrick family (*Chatterjee, 1985*), the species epithet should be *Postosuchus kirkpatrickorum*; however, an emendation was never made and the present version (4th Edition) of the International Code of Zoological Nomenclature no longer requires that such emendations be made (*ICZN, 1999*).

*Key References* – *Chatterjee (1985)*; *Nesbitt (2011)*; *Weinbaum (2011)*; *Weinbaum (2013)*.

### *Redondasuchus rinehardti* (*Spielmann et al., 2006*)

*Holotype* – NMMNH P-43312, partial right trunk paramedian osteoderm (*Spielmann et al., 2006*).

*Referred Material* – see *Spielmann et al. (2006)*. With permission, unpublished material currently under study by Jeffrey Martz and Axel Hungerbühler at Mesalands Dinosaur Museum in Tucumcari, New Mexico is also scored.

*Age* – Late Triassic, late Norian to Rhaetian, Apachean (*Spielmann & Lucas, 2012*).

*Occurrence* – Redonda Formation, Dockum Group, New Mexico, U.S.A (*Spielmann & Lucas, 2012*).

*Remarks* – A fair amount of aetosaurian osteoderm material has been recovered from the Upper Triassic Redonda Formation of New Mexico, most of which appears to be from at least one typothoracine. *Redondasuchus reseri* was named by *Hunt & Lucas (1991)* for a small typothoracine aetosaurs that reportedly lacked lateral osteoderms, and instead proposed a novel reconstruction for an aetosaurian in which the flexed outer edge of the trunk paramedians covered the flank of the animal rather than a separate laterally situated osteoderm (*Heckert, Hunt & Lucas, 1996*). However, the holotype trunk osteoderm was interpreted backwards by those authors with the flexed 'outer edge' actually being situated along the midline of the carapace. Moreover, there is no direct evidence that *Redondasuchus reseri* differed from all other aetosaurs in lacking lateral osteoderms (*Martz, 2002*).

*Martz (2002)* could not distinguish the osteoderms of *Redondasuchus reseri* from those of *Typothorax coccinarum* in any characteristic other than size, but *Spielmann et al. (2006)* argued that *Redondasuchus reseri* was indeed distinct and named a second species, *Redondasuchus rineharti*, for isolated osteoderms and a proximal femur head from a larger aetosaurian. Those authors differentiated the new species from *Redondasuchus reseri* based on larger size and the presence of a dorsal eminence on the paramedian osteoderms. Differentiation based on size is problematic as no ontogenetic study has been made for *Redondasuchus* to refute the idea that the holotype and referred specimens of *Redondasuchus reseri* are simply skeletally immature specimens of another typothoracine. Moreover, in *Typothorax coccinarum*, the more anterior trunk paramedian osteoderms lack dorsal eminences. Furthermore, strong flexion of paramedian osteoderms occurs in several aetosaur taxa including *Typothorax coccinarum* (PEFO 23388), *Paratypothorax* sp. (PEFO 3004), *Sierritasuchus macalpini* (UMMP V60817), and *Calyptosuchus wellesi* (UCMP 136744). Thus, *Redondasuchus reseri* lacks clear autapomorphies or even a unique combination of characters and it is not included in this study pending future reexamination. However, there are some fundamental differences between *Redondasuchus rineharti* and *Typothorax coccinarum* including the more closely packed and deep pits in *Redondasuchus rineharti*, as well as the oblong pits in the transverse trough posterior to the anterior bar and it is included in the present analysis, supplemented by scorings from new undescribed material from New Mexico (J. Martz, personal communication, 2013).

*Key References* – *Spielmann et al. (2006)* & *Spielmann & Lucas (2012)*.

### *Revueltosaurus callenderi* (*Hunt, 1989*)

*Holotype* – NMMNH P-4957, nearly complete premaxillary tooth.

*Referred Material* – PEFO 33787, partial skeleton and skull; PEFO 33788, partial skull; PEFO 34269, partial skeleton and skull; PEFO 34561, nearly complete skeleton and skull; PEFO 36875, nearly complete skeleton and skull; PEFO 36876, partial skeleton and skull (*Parker & Martz, 2011*; *Nesbitt, 2011*; *Parker et al., 2005*, *Parker et al., 2007*).

*Age* – Late Triassic, mid to late Norian, Revueltian (*Ramezani et al., 2011*; *Parker & Martz, 2011*).

*Occurrence* – Petrified Forest Member, Chinle Formation, Arizona, U.S.A.; Bull Canyon Formation, Dockum Group, New Mexico, U.S.A (*Hunt, 1989*; *Parker et al., 2005*).

*Remarks* – Originally known from only isolated teeth that were assigned to ornithischian dinosaurs (*Hunt, 1989*; *Padian, 1990*; *Heckert, 2003*), *Revueltosaurus callenderi* is currently one of the most completely documented pseudosuchians based on an as of yet undescribed series of skeletons recovered from the Chinle Formation of Petrified Forest National Park in Arizona (*Parker et al., 2005*, *Parker et al., 2007*; *Nesbitt, 2011*; *Farlow et al., 2014*). A current phylogenetic analysis of the Archosauriformes recovers *Revueltosaurus callenderi* as the sister taxon of Aetosauria (*Nesbitt, 2011*).

*Key References* – *Heckert (2003)*; *Parker et al. (2005)*; *Parker et al. (2007)*; *Nesbitt (2011)*.

### *Rioarribasuchus chamaensis* (*Zeigler, Heckert & Lucas, 2003*)

*Holotype* – NMMNH P-32793, right anterior caudal paramedian osteoderm (*Zeigler, Heckert & Lucas, 2003*).

*Referred Material* – see *Parker (2007)*.

*Age* – Late Triassic, mid-late Norian, Revueltian (*Irmis et al., 2011*).

*Occurrence* – Petrified Forest Member, Chinle Formation, New Mexico, U.S.A.; Martha's Butte beds, Sonsela Member, Chinle Formation, Arizona, U.S.A (*Zeigler, Heckert & Lucas, 2003*; *Parker & Martz, 2011*).

*Remarks* – *Rioarribasuchus chamaensis* was first described as a new species of *Desmatosuchus* by *Zeigler, Heckert & Lucas (2003)* based on isolated paramedian and lateral osteoderms from the Revueltian Snyder Quarry in New Mexico. *Parker (2003)* demonstrated with a phylogenetic analysis that "*Desmatosuchus*" *chamaensis* was closer to *Paratypothorax* rather than *Desmatosuchus*, a finding opposed by *Heckert, Zeigler & Lucas (2003)* who argued that the taxon was more like *Desmatosuchus* than *Paratypothorax*. *Parker & Irmis (2005)* & *Parker (2006)* reiterated that "*Desmatosuchus*" *chamaensis* should be assigned to a new genus, differing from studies such as *Lucas et al. (2005)*; *Heckert, Lucas & Hunt (2005)*; who continued to assign the species to the genus *Desmatosuchus*. Subsequently two names were coined for the taxon nearly simultaneously, *Heliocanthus* (*Parker, 2007*) and *Rioarribasuchus* (*Lucas, Hunt & Spielmann, 2006*); however, the paper by *Lucas, Hunt & Spielmann (2006)* was published earlier and thus the name *Rioarribasuchus* has priority. The status of the taxonomic naming was considered controversial (e.g., *Dalton, 2008*), but was resolved by *Irmis et al. (2007)*, who as first reviser, used the name *Rioarribasuchus chamaensis* and accordingly *Heliocanthus* is a junior objective synonym of *Rioarribasuchus*. The close relationship between *Rioarribasuchus* and *Paratypothorax* has been recovered by all current phylogenetic analyses of the Aetosauria (*Parker, 2007*; *Desojo, Ezcurra & Kischlat, 2012*; *Heckert et al., 2015*). Indeed *Rioarribasuchus chamaensis* possesses no desmatosuchine apomorphies (*Parker, 2007*); however, some workers still consider *Rioarribasuchus* to be a desmatosuchine (e.g., *Lichtig & Lucas, 2014*) although they have not supported this with a phylogenetic analysis.

*Parker (2007)* also provided a novel reconstruction of *Rioarribasuchus chamaensis* in which the sacral and anterior caudal paramedian osteoderms possess dorsal eminences that have the form of an elongate, anterior medially directed, curved spine. The presence of these eminences is an autapomorphy of the taxon. The orientation and placement of the osteoderms with the spines was criticized by *Lucas & Connealy (2008)* & *Lichtig & Lucas (2014)*; however, orientation of the osteoderms using the anterior bar and the direction of osteoderm edge tapering demonstrates that the orientation proposed by *Parker (2007)* must be correct. The anterior paramedians and all of the lateral osteoderms are identical to *Paratypothorax*, and originally were thought to represent that taxon by the discoverers (*Heckert & Zeigler, 2003*). The presence of lateral plates identical to paratypothoracin aetosaurians supports the hypothesis that the osteoderms with the elongate recurved spines must be paramedians and not laterals (*Parker, 2007*; differing from *Lucas & Connealy, 2008*; *Lichtig & Lucas, 2014*). *Rioarribasuchus chamaensis* is currently known from the Snyder and Hayden quarries in the Chama Basin of New Mexico and from Petrified Forest National Park in Arizona. All three of these localities are in Revueltian strata of the Chinle Formation (*Heckert, Lucas & Hunt, 2005*; *Irmis et al., 2007*; *Parker & Martz, 2011*).

*Rioarribasuchus chamaensis* is currently known mainly from osteoderms, although *Heckert, Zeigler & Lucas (2003)* referred two astragali (NMMNH P-33927, NMMNH P-33932) and a calcaneum (NMMNH P-33931) from the Snyder Quarry. Those authors did not list any apomorphies or provide any comparisons to other taxa for the astragali and thus this referral is ambiguous given the co-occurrence of *Typothorax coccinarum* in the quarry. However, they did note that the referred calcaneum is not as dorsoventrally compressed as the calcaneum of *Typothorax coccinarum* (presumably AMNH FR 2713). Unfortunately there are no recognized paratypothoracin distal tarsals to use for a comparison to help verify these assignments. An isolated anterior aetosaurian caudal vertebrae (GR 174) from the Hayden Quarry bears caudal ribs that originate close to the base of the centrum rather than at the base of the neural arch. This character only occurs in *Paratypothorax* sp. (PEFO 3004) and not in *Typothorax* (*Martz, 2002*) so the Hayden Quarry vertebra is assigned to Paratypothoracini, most likely *Rioarribasuchus chamaensis*.

*Key References* – *Zeigler, Heckert & Lucas (2003)*; *Heckert, Zeigler & Lucas (2003)*; *Parker (2007)*.

### *Scutarx deltatylus gen. et sp. nov.*

*Holotype* – PEFO 34616, partial skull, cervical paramedian and lateral osteoderms.

*Zoobank LSID* – urn:lsid:zoobank.org:act:E06A8E11-5864-4717-AFA2-9021842B886D

*Referred Material* – PEFO 31217, much of a postcranial skeleton including vertebrae, ribs, pectoral and pelvic girdles, osteoderms; PEFO 34919, much of a postcranial skeleton including vertebrae, ribs, osteoderms, ilium; PEFO 34045, much of a postcranial skeleton including vertebrae, ribs, and osteoderms; TTU P-09420, two paramedian osteoderms; UCMP 36656, paramedian and lateral osteoderms. The last two specimens were previously referred to *Calyptosuchus wellesi* (*Martz et al., 2013*).

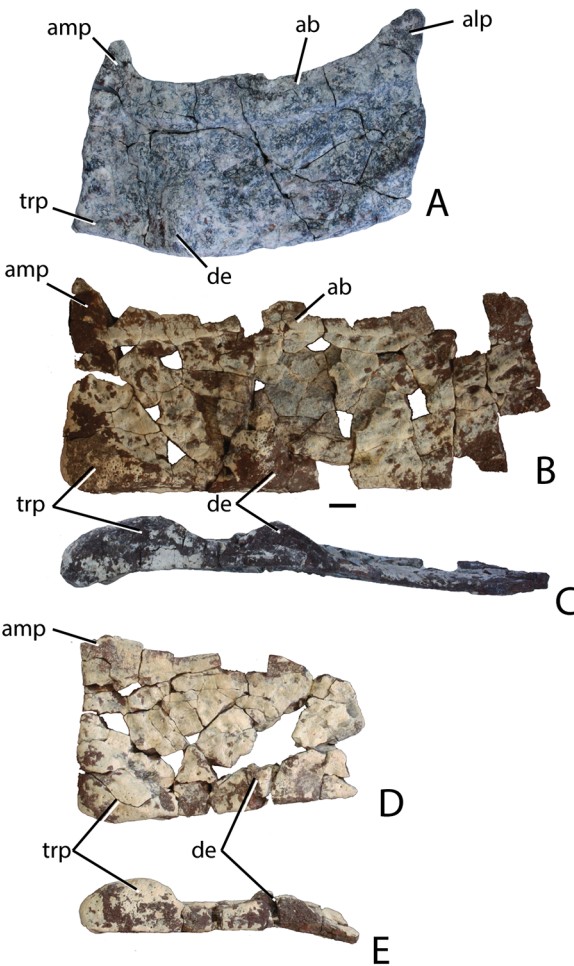

**Figure 2 Holotype paramedian osteoderms of *Scutarx deltatylus* from PEFO 34616.** (A) posterior cervical osteoderm in trunk view. (B and C) right trunk paramedian osteoderm in dorsal (B) and posterior (C) views. (D and E) partial right trunk paramedian osteoderm in dorsal (D) and posterior (E) views. Note the prominence of the triangular protuberance in the posterior views. Scale bar equals 1 cm. Abbreviations: **ab**, anterior bar; **alp**, anterolateral process; **amp**, anteromedial process; **de**, dorsal eminence; **trp**, triangular protuberance.

*Age* – Late Triassic, middle Norian, late Adamanian (*Ramezani et al., 2011*; *Parker & Martz, 2011*; *Martz et al., 2013*).

*Occurrence* – lower part of the Sonsela Member, Chinle Formation, Arizona, U.S.A.; middle part of the Cooper Canyon Formation, Dockum Group, Texas, U.S.A (*Parker & Martz, 2011*).

*Etymology* – *Scutarx* 'shield fortress,' from Latin *scutum* 'shield' + Latin *arx* 'fortress, castle;' *deltatylus* 'triangular protuberance,' from Greek delta + Greek tylos 'knob, knot, swelling, callous, protuberance.'

*Diagnosis* – Medium-sized aetosaurian diagnosed by the following autapomorphies; the cervical and trunk paramedian osteoderms bear a strongly raised, triangular tuberosity in the posteromedial corner of the dorsal surface of the osteoderm (Fig. 2); the occipital condyle lacks a distinct neck because the condylar stalk is mediolaterally broad

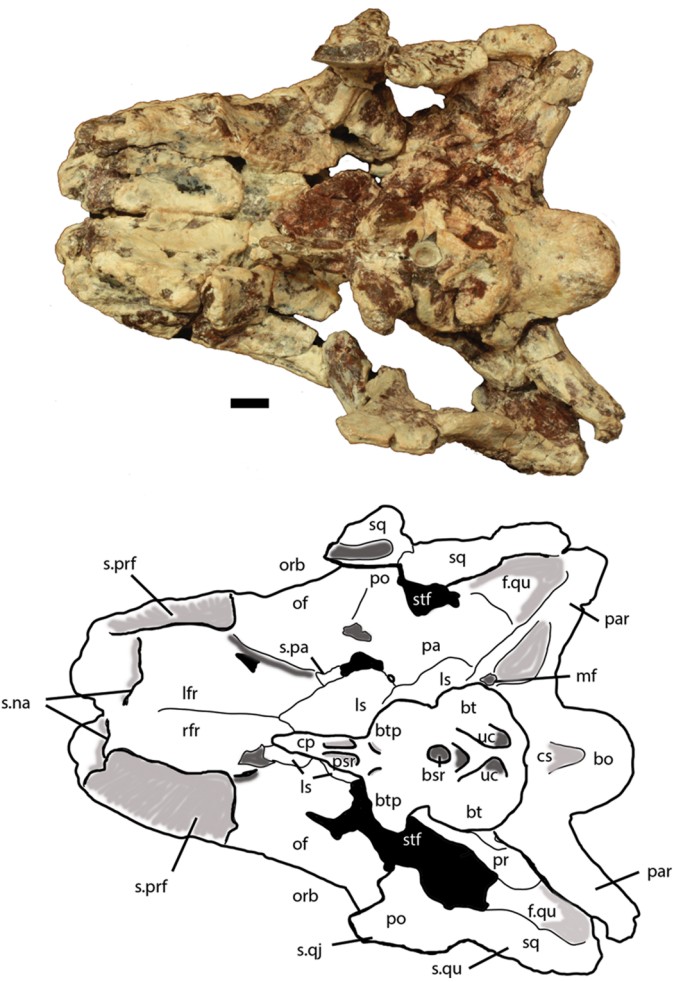

**Figure 3 Holotype skull of *Scutarx deltatylus* (PEFO 34616) in ventral view.** Scale bar equals 1 cm. Abbreviations: **bpt**, basipterygoid processes; **bsr**, basisphenoid recess; **bt**, basal tubera; **cp**, cultriform process; **crp**, crista prootica; **f.**, fossa for specified element; **lfr**, left frontal; **lr**, lateral ridge; **ls**, laterosphenoid; **of**, orbital fossa; **orb**, orbit; **par**, paroccipital process of the opisthotic; **po**, postorbital; **prf**, prefrontal; **pr**, prootic; **prf**, prefrontal; **psr**, parasphenoid recess; quadrate; **rfr**, right frontal; **sq**, squamosal; **ssr**, subsellar recess; **stf**, supratemporal fenestra; **uc**, unossified cleft of the basal tubera.

(Fig. 3); the base of the cultriform process of the parabasisphenoid bears deep lateral fossae (parasphenoid recesses; Figs. 3 and 4); the frontals and parietals are very thick dorsoventrally; and there is a distinct fossa or recess on the lateral surface of the ilium between the supraacetabular crest and the posterior portion of the iliac blade (Fig. 5). *Scutarx deltatylus* can also be differentiated from other aetosaurs by a unique combination of characters including moderately wide trunk paramedian osteoderms with a strongly raised anterior bar that possesses anteromedial and anterolateral processes (shared with all aetosaurians except Desmatosuchini; Fig. 2); osteoderm surface ornamentation of radiating ridges and pits that emanate from a posterior margin contacting a dorsal eminence (shared with *Calyptosuchus wellesi*, *Stagonolepis robertsoni*, *Adamanasuchus eisenhardtae*, *Neoaetosauroides engaeus*, *Aetobarbakinoides brasiliensis*, and *Aetosauroides scagliai*); lateral trunk osteoderms with an obtuse angle between the dorsal and lateral

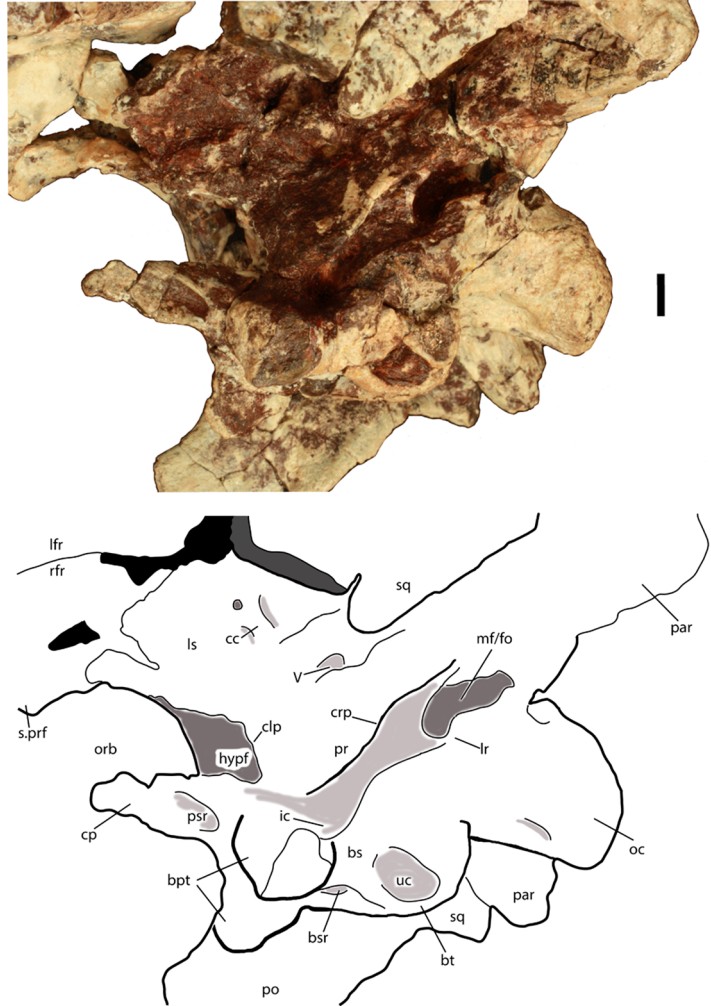

**Figure 4 Braincase of *Scutarx deltatylus* (PEFO 34616) in ventrolateral view.** Scale bar equals 1 cm. Abbreviations: **bpt**, basipterygoid processes; **bsr**, basisphenoid recess; **bt**, basal tubera; **cc**, cotylar crest; **clp**, clinoid process; **cp**, cultriform process; **crp**, crista prootica; **fo**, foramen ovale; **hypf**, hypophyseal fossa; **ic**, exit area of the internal carotid artery; **lfr**, left frontal; **lr**, lateral ridge; **ls**, laterosphenoid; **mf**, metotic foramen; **na**, nasal; **oc**, occipital condyle; **orb**, orbit; **pa**, parietal; **par**, paroccipital process of the opisthotic; **po**, postorbital; **pr**, prootic; **prf**, prefrontal; **psr**, parasphenoid recess; **rfr**, right frontal; **s.**, suture with designated element; **sq**, squamosal; **uc**, unossified cleft of the basal tubera; **V**, passageway for the Trigeminal nerve (CN V).

flanges (shared with non-desmatosuchines); a dorsoventrally short pubic apron with two proximally located 'obturator' fenestrae (shared with *Stagonolepis robertsoni*); and an extremely anteroposteriorly short parabasisphenoid, with basal tubera and basipterygoid processes almost in contact and a reduced cultriform process (Fig. 3; shared with *Desmatosuchus*).

*Remarks* — Aetosaurian material referable to *Calyptosuchus* occurs through Adamanian-age deposits in Arizona, New Mexico, and Texas. In Arizona, specimens from the Sonsela Member previously referred to *Calyptosuchus wellesi* (e.g., *Long & Murry, 1995*; *Parker & Irmis, 2005*; *Parker, 2005a*; *Parker, 2006*; *Parker & Martz, 2011*; *Martz et al., 2013*)

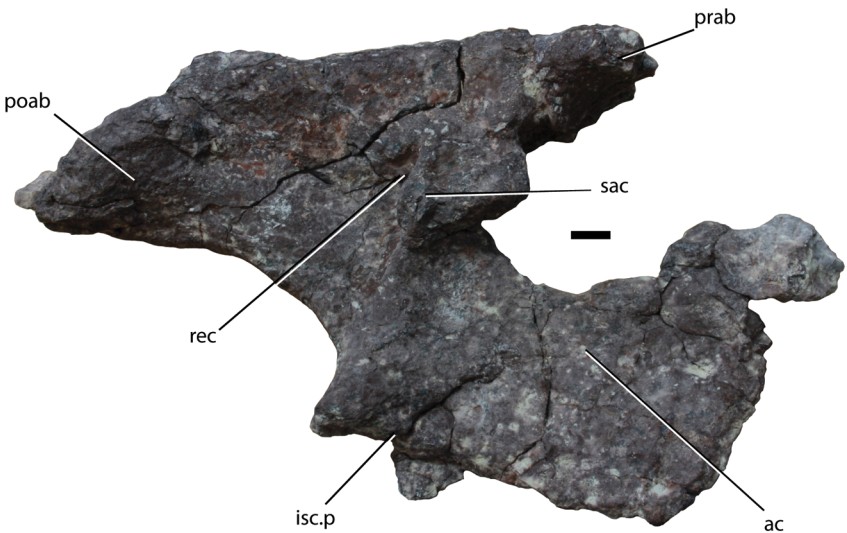

**Figure 5 Right ilium of *Scutarx deltatylus* (PEFO 34919) in 'lateral' view (see text for discussion regarding anatomical direction of the ilium).** Scale bars equal 1 cm. Abbreviations: **ac**, acetabulum; **isc.p**, ischiadic peduncle; **poab**, posterior process of the iliac blade; **prab**, anterior process of the iliac blade; **rec**, recess; **sac**, supraacetabular crest.

possess a distinctive raised triangular boss on the posteromedial corner of the dorsal surface of the paramedian osteoderms (Fig. 2). Detailed comparison demonstrates that this character is not present in the holotype of *Calyptosuchus wellesi* (UMMP 13950) or in referred material of that taxon from the *Placerias* Quarry. Thus, this feature is autapomorphic of a new taxon, *Scutarx deltatylus,* which is named and diagnosed here, but will be fully described elsewhere. In this analysis, *Scutarx deltatylus* is coded from four new, partial skeletons from Petrified Forest National Park in Arizona. Newly recognized osteoderms of *Calyptosuchus* (TTU P-09420) from the Post Quarry of Texas also possess the diagnostic triangular boss and thus are actually referable to *Scutarx deltatylus* and not *Calyptosuchus wellesi* (differing from the interpretation by *Martz et al. (2013)*). This occurrence supports correlation of the Post Quarry (middle Cooper Canyon Formation) to the lower part of the Sonsela Member of Arizona as suggested by *Martz et al. (2013)*. Thus it may be possible to subdivide the Adamanian biozone utilizing *Calyptosuchus* and *Scutarx.*

   *Key References* – *Parker (2014)*; *Parker & Irmis (2005)*; *Martz et al. (2013)*.

### *Sierritasuchus macalpini* (*Parker, Stocker & Irmis, 2008*)

*Holotype* – UMMP V60817, partial postcranial skeleton consisting of vertebrae and osteoderms (*Parker, Stocker & Irmis, 2008*).

   *Referred Material* – TTU P-10731, left lateral osteoderm.

   *Age* – Late Triassic, early to mid-Norian, Adamanian (*Ramezani et al., 2011*; *Lucas, 2010*).

   *Occurrence* – Tecovas Formation, Dockum Group, Texas, U.S.A (*Long & Murry, 1995*; *Parker, Stocker & Irmis, 2008*).

*Remarks* – The holotype (UMMP V60817) of *Sierritasuchus macalpini* was collected in 1939 from the Tecovas Formation of Texas by the late Archie J. MacAlpin (University of Notre Dame), who at the time was a student of Ermine C. Case of the University of Michigan. The specimen, which consists of vertebrae and osteoderms from the cervical and trunk regions, was originally referred to *Desmatosuchus haplocerus* by *Long & Murry (1995)*. *Parker (2002)* questioned this referral and considered the possibility that UMMP V60817 represented a skeletally immature specimen of *Longosuchus meadei* even though it was from a higher stratigraphic position.

Redescribed by *Parker, Stocker & Irmis (2008)*, this was the first aetosaurian specimen to have osteoderms histologically sampled to help determine the ontogenetic stage of the specimen. Histological analysis suggested that although it is not a full grown adult, the preserved material has no indicators of skeletal immaturity either (*Parker, Stocker & Irmis, 2008*). Within Desmatosuchinae *Sierritasuchus macalpini* shares more characters with *Longosuchus meadei* than *Desmatosuchus spurensis*, but differs from the former in possessing dorsoventrally flattened, non- faceted, recurved spines on the lateral osteoderms. *Parker, Stocker & Irmis (2008)* listed an additional difference, the lack of radial pattern on the dorsal paramedian osteoderms, but subsequent examination of the type materials of *Longosuchus meadei* demonstrate a random not radial pattern (*Parker & Martz, 2010*). *Longosuchus meadei* was scored as having a radial pattern in existing phylogenetic analyses (*Heckert, Hunt & Lucas, 1996*; *Heckert & Lucas, 1999*), and this scoring was repeated in subsequent analyses (*Parker, 2007*; *Parker, Stocker & Irmis, 2008*). Determining the exact position of *Sierritasuchus macalpini* within Desmatosuchinae has been problematic (*Parker, 2007*; *Parker, Stocker & Irmis, 2008*), but *Desojo, Ezcurra & Kischlat (2012)* recovered *Sierritasuchus macalpini* as the earliest branching member of the Desmatosuchinae.

*Key References* – *Parker, Stocker & Irmis (2008)* & *Desojo, Ezcurra & Kischlat (2012)*.

## Unnamed taxon SMNS 19003

*Age* – Late Triassic, Norian, Revueltian (*Deutsche Stratigraphische Kommission, 2005*; *Lucas, 2010*).

*Occurrence* – Lower and middle Stubensandstein, Löwenstein Formation, Germany (*Desojo et al., 2013*).

*Remarks* – SMNS 19003 represents an almost complete, articulated skeleton of a paratypothoracin aetosaur from the Upper Triassic of Germany. The specimen includes a beautifully preserved skull, which is the only unambiguous, non-braincase skull material known for a paratypothoracin. *Desojo et al. (2013)* refer the specimen as *Paratypothorax andressorum*, but the material has yet to be described and is currently under study by Rainer Schoch and Julia Desojo (*Desojo & Schoch, 2014*). However, some details of the skull were presented by *Sulej (2010)*. One notable characteristic of the skull is that the apex of the premaxilla lacks the transverse expansion found in aetosaurs such as *Desmatosuchus* and *Stagonolepis* (*Desojo & Schoch, 2014*). *Typothorax coccinarum* (PEFO 38001/YPM 58121) also lacks this expansion, suggesting that this may be an apomorphy for Typothoracinae.

*Key References* – *Sulej (2010)*; *Desojo et al. (2013)*; *Desojo & Schoch (2014)*.

### *Stagonolepis robertsoni* (*Agassiz, 1844*)

*Holotype* – EM 27 R, impression of a segment of the plastron (*Agassiz, 1844*).

*Referred Material* – see *Walker (1961)* for a full list; particularly important is MCZD 2, an articulated partial skeleton including much of the skull with a well preserved braincase and articulated nuchal and cervical paramedian osteoderms.

*Age* – Late Triassic, ?Carnian (*Lucas, 2010*).

*Occurrence* – Lossiemouth Sandstone Formation, Scotland, U.K (*Walker, 1961*).

*Remarks* – Originally described by *Agassiz (1844)* as a fish from what was thought to be the Old Red Sandstone in Scotland, Charles Lyell first raised suspicions that *Stagonolepis* might instead be a reptile more closely related to *Mystriosuchu*s (*Huxley, 1859*). Reexamination the material showed it to be a parasuchian reptile and provided the first solid evidence that the Lossiemouth Sandstone Formation was Triassic in age (*Huxley, 1859*; *Huxley, 1875*; *Huxley, 1877*). Unfortunately much of the collected material consists of natural molds, which has made study of the specimens difficult and only possible through the making of casts (*Huxley, 1859*; *Huxley, 1877*). *Stagonolepis robertsoni* was fully described by *Walker (1961)* who developed a new technique of creating flexible PVC casts to recover additional details from the deeper portions of the molds than was available to Huxley. *Walker (1961)* also had the benefit of new specimens, most importantly an actually articulated body fossil (MCZD 2), which represents a nearly complete skull and the anterior cervical armor (*Walker, 1961*; *Gower & Walker, 2002*). This specimen allowed for detailed reconstruction of the skull and braincase and demonstrated clearly that *Stagonolepis robertsoni* was an aetosaurian rather than a phytosaur as previously believed (e.g., *Camp, 1930*).

Although *Walker's (1961)* reconstruction of *Stagonolepis robertsoni* relied significantly on observations made from *Aetosaurus ferratus*, examination of the MCZD specimen and the NHMUK casts show that Walker's work is extremely reliable for comparisons; however, character scorings for this analysis are taken from the fossils and casts, not from the published reconstruction. This, of course, is based on the assumption that only a single taxon is present in the Scottish quarries. Walker did note the presence of two different size categories in the specimens, but determined any anatomical differences between the two to represent sexual dimorphism. There is currently no evidence to refute this hypothesis, the most notable difference is in the coverage of ornamentation on the dorsal paramedian osteoderms where in the smaller individuals the posterior portions of the dorsal surfaces are devoid of any ornamentation. Unfortunately all of the quarries where all of the *Stagonolepis robertsoni* material originates have been closed and grown over, and it is unlikely that more material of *Stagonolepis robertsoni* will be found in the immediate future.

What is clear from examination of the Scottish material is that *Stagonolepis robertsoni* is anatomically distinct from *Calyptosuchus* from North America, and *Aetosauroides scagliai* from South America (*Parker & Martz, 2011*; *Desojo & Ezcurra, 2011*; differing from *Lucas & Heckert, 2001*; *Heckert & Lucas, 2002*). Although all share a basic radial patterning and a medially offset dorsal eminence, there are key differences in the osteoderms and

especially in the cranial material of these taxa. Therefore, all three are treated as separate terminal taxa in this analysis.

Key References – *Huxley (1877)*; *Walker (1961)*; *Gower & Walker (2002)*.

### *Stagonolepis olenkae* (*Sulej, 2010*)

*Holotype* –ZPAL AbIII/466/17, skull roof (*Sulej, 2010*).

*Referred Material* – see *Sulej (2010)*.

*Age* – Late Triassic, late Carnian (*Dzik & Sulej, 2007*).

*Occurrence* – Drawno beds, Krasiejów, Opole Silesia, Poland (*Sulej, 2010*).

*Remarks* – *Stagonolepis olenkae* was described by *Sulej (2010)* for remarkably well preserved aetosaur material from the Krasiejów quarry in Poland (*Dzik, 2001*; *Dzik & Sulej, 2007*). The original description of the holotype (*Sulej, 2010*) is based mainly on the skull material; unfortunately much of the descriptive text is identical to that of *Walker (1961)* so it is not clear if the Polish material is accurately described. *Sulej (2010)* provides some obscure references to postcranial material (e.g., mentioning of a tibia in the diagnosis), but other than some of this material being mentioned and partly figured by *Dzik (2001)* & *Lucas, Spielmann & Hunt (2007)* have provided the only descriptions and photographs of this material, but assigned it to *Stagonolepis robertsoni* based mainly on the ornamentation of the trunk paramedian osteoderms. The most recent discussion of this material suggests that characters used to differentiate *Stagonolepis olenkae* from *Stagonolepis robertsoni* are polymorphic and *S. olenkae* is simply a variant *of S. robertsoni* (*Antczak, 2015*). This hypothesis is tested phylogenetically here for the first time.

Key References – *Sulej (2010)*; *Lucas, Spielmann & Hunt (2007)*; *Antczak (2015)*.

### *Stenomyti huangae* (*Small & Martz, 2013*)

*Holotype* – DMNH 60708, skull with lower jaws, partial postcranial skeleton including a well-preserved plastron (*Small & Martz, 2013*).

*Referred Material* – DMNH 61392, partial skull with lower jaws, osteoderms, ribs, and vertebrae; DMNH 34565, maxilla, scapula, pubis, ribs and osteoderms.

*Age* – Late Triassic, middle – late Norian, Revueltian (*Ramezani et al., 2011*; *Small & Martz, 2013*).

*Occurrence* – red siltstone member, Chinle Formation, Eagle County, Colorado (*Small & Martz, 2013*).

*Remarks* – *Stenomyti huangae* is a well-documented small aetosaurian that, when originally discovered, was presented as the first good evidence for the presence of *Aetosaurus* in western North America (*Small, 1998*). Subsequent preparation and study revealed that it was anatomically distinct (*Small & Martz, 2013*). *Stenomyti huangae* possesses a unique ventral armor arrangement, which instead of rows of articulated square osteoderms, consists of an arrangement of oval and irregularly shaped osteoderms that do not contact each other. The removal of these specimens from the genus *Aetosaurus* eliminates a proposed biochronological correlation between Europe and eastern North America, with western North America (*Lucas, Heckert & Huber, 1998*).

Key References – *Small (1998)* & *Small & Martz (2013)*.

### Tecovasuchus chatterjeei (*Martz & Small, 2006*)

*Holotype* – TTU P-00545, paramedian and lateral osteoderms of the trunk region, braincase, partial vertebra (*Martz & Small, 2006*).

*Referred Material* – UMMP 9600, right trunk paramedian osteoderm; TTU P-09222, left trunk paramedian osteoderm; TTU P-07244, trunk paramedian osteoderm; NMMNH P-25641, left (?) trunk lateral osteoderm; TMM 31173-54, partial left paramedian osteoderm; PEFO 37871, partial paramedian osteoderm; MNA V3202, partial right paramedian osteoderm, three right trunk lateral osteoderms, one ?left trunk lateral osteoderm fragment (*Parker, 2005a*); MNA V3000, left trunk lateral osteoderm; MNA V2898, left trunk lateral osteoderm (*Heckert et al., 2007*).

*Age* – Late Triassic, early to middle Norian, Adamanian (*Lucas, 2010*).

*Occurrence* – Tecovas Formation, Dockum Group, Texas, U.S.A.; ?Bluewater Creek Member, Chinle Formation, New Mexico, U.S.A.; upper Blue Mesa Member, Chinle Formation, Arizona, U.S.A (*Parker, 2005a*; *Martz & Small, 2006*; *Heckert et al., 2007*).

*Remarks* – The holotype (TTU P-00545) was collected in the 1950s by Wann Langston Jr. from the Tecovas Formation near Potter County, Texas. A referred specimen (UMMP 9600) was collected near the same area in 1925 by William Buettner of the University of Michigan. TTU P-00545 was assigned to *Typothorax coccinarum* by *Small (1985*:8*)* and TTU P-00545, TTU P-09222, and UMMP 9600 were assigned to *Paratypothorax* sp. by *Long & Murry (1995)*. *Lucas, Heckert & Hunt (1995)* recognized the distinctness of the UMMP osteoderm, but hesitated to erect a new taxonomic name based on a single osteoderm and were apparently unaware of the Texas Tech specimen. The TTU material was later described under the name *Tecovasuchus chatterjeei* (*Martz & Small, 2006*).

*Parker (2005a)* & *Heckert et al. (2007)* referred material from the lower part of the Chinle Formation to *Tecovasuchus*, including MNA V3202, which had previously used as support for the presence of cervical spines in *Calyptosuchus wellesi* (*Long & Ballew, 1985*; *Long & Murry, 1995*). The lateral osteoderms of MNA V3202 possess apomorphies of Paratypothoracini most notably the greatly reduced triangular dorsal flange. The preserved paramedian osteoderm in MNA V3202 appears to have a high width/length ratio and the posterior edge is distinctly beveled, which is an autapomorphy of *Tecovasuchus chatterjeei* (*Parker, 2005a*; *Martz & Small, 2006*). PEFO 37871 is a portion of a paramedian osteoderm that also preserves the beveled posterior edge and therefore represents another occurrence from the lower part of the Chinle Formation, in this case the upper Blue Mesa Member. *Tecovasuchus chatterjeei* has been postulated as an index taxon for the early Adamanian (*Heckert et al., 2007*). These authors also assigned additional material from the NMMNH collections (*Heckert et al., 2007*:Fig. 3) to *Tecovasuchus*; however, no apomorphies of that taxon are apparent in the published figures or listed in the text so those specimens are not included here.

*Key References* – *Lucas, Heckert & Hunt (1995)*; *Parker (2005a)*; *Martz & Small (2006)*; *Heckert et al. (2007)*.

### Typothorax coccinarum (*Cope, 1875*)

*Lectotype* – USNM 2585, five paramedian osteoderm fragments.

*Referred Material* – Numerous specimens, see *Long & Murry (1995)*; *Hunt (2001)*; *Martz (2002)*; & *Parker & Martz (2011)* for lists. Notable referred specimens include AMNH FR 2709, paramedian osteoderms, left femur; AMNH FR 2710, right femur (probably same specimen as AMNH FR 2709); AMNH FR 2713, lateral osteoderms, right femur, left calcaneum, caudal vertebra (lectotype of *Episcoposaurus horridus*); NMMNH P- 56299, articulated carapace missing the skull; NMMNH P-12964, nearly complete skeleton with skull (mostly destroyed); TTU P-09214, osteoderms, vertebrae, braincase, dentary; UCMP 34227, numerous trunk paramedian osteoderms; UCMP 34255, articulated tail, limb and girdle material; PEFO 38001/YPM 58121, associated skeleton with complete skull; partial skeleton with skull (still in preparation).

*Age* – Late Triassic, middle to late Norian, latest Adamanian and Revueltian (*Ramezani et al., 2011*; *Irmis et al., 2011*).

*Occurrence* – Sonsela and Petrified Forest members, Chinle Formation, Arizona, U.S.A.; middle part of the Cooper Canyon Formation, Dockum Group, Texas, U.S.A.; Bull Canyon Formation, Dockum Group, New Mexico, U.S.A. (*Long & Ballew, 1985*; *Heckert et al., 2010*; *Parker & Martz, 2011*; *Martz et al., 2013*).

*Remarks* – Fossils of *Typothorax coccinarum* are extremely common in Revueltian rocks across the southwestern United States, but despite the large amount of available material most specimens have only been superficially or not described. An exception is a nearly complete skeleton (NMMNH P-56299) described by *Heckert et al. (2010)*, which provides key information on the lateral osteoderms and especially the ventral armor. Some of the best figured materials are from the Canjilon Quarry (*Martz, 2002*), which forms the basis of much of the description by *Long & Murry (1995)* as well as our understanding of the taxon.

Until recently the only known cranial material was a complete skull (NMMNH P-12964) from the Bull Canyon Formation (Dockum Group) of New Mexico. This skull was very preliminarily described by *Hunt, Lucas & Reser (1993)* and later figured, but not described by *Heckert et al. (2010)*. Unfortunately this specimen was badly damaged during molding and is currently only visible in a cast (NMMNH C-4638) that is on exhibit at the New Mexico Museum of Natural History and Science (*Heckert et al., 2010*:628). Fieldwork by Yale University in the Petrified Forest Member (Chinle Formation) of Petrified Forest National Park in the summer of 2008 resulted in the discovery of two skeletons of *Typothorax coccinarum* both which include well-preserved skulls. One of these skulls (PEFO 38001/YPM 58121) was used to code *Typothorax* for this study, but unfortunately the braincase is not exposed in that specimen. The second skull is still in preparation (M. Fox, personal communication, 2014).

The type material of *Typothorax coccinarum* consists of only a few fragments of paramedian osteoderms and most descriptions and referrals have been made using better preserved material such as AMNH FR 2709, AMNH FR 2710, or UCMP 34227. The type material is not diagnostic above the level of Typothoracinae and accordingly *Typothorax coccinarum* is most likely a *nomen dubium* (*Parker, 2013*).

Note that, following discussion by *Parker (2006)*; *Parker & Martz (2011)* & *Martz et al. (2013)* *Typothorax antiquum* (*Lucas, Heckert & Hunt, 2003*) is not considered to be a valid

taxon in this study, but rather a less skeletally mature specimen of *Typothorax coccinarum*. The occurrence (NMMNH P-25745) of the Revueltian index taxon *Machaeroprosopus* (=*Pseudopalatus*) at the type locality for *Typothorax antiquum* also necessitates a detailed review of the stratigraphic position of this material, which is purportedly Adamanian in age (*Lucas, Heckert & Hunt, 2003*; *Hunt, Lucas & Heckert, 2005*).

*Key References* – *Cope (1875)*; *Cope (1877)*; *Cope (1887)*; *Long & Ballew (1985)*; *Long & Murry (1995)*; *Martz (2002)*; *Heckert et al. (2010)*; *Parker (2013)*.

## PHYLOGENETIC ANALYSIS

The character matrix of 28 taxa and 83 characters (Appendices A and B) was assembled and edited in Morphobank (*O'Leary & Kaufman, 2012*) as matrix number 2617 of project number 1009, and exported as a NEXUS file (Appendix A). Submatrices (partitions) were edited using NEXUS Data Editor for Windows version 5.0 (*Page, 2001*). All matrices were analyzed in PAUP* (Version 4.0b10 for 32-bit Microsoft Windows, *Swofford, 2003*). *Postosuchus kirkpatricki* was constrained as the outgroup for the analysis. *Revueltosaurus callenderi* was utilized as a second outgroup, but unconstrained.

PAUP* determined three characters to be parsimony uninformative (39, 42, 72), which were excluded *a priori* to eliminate inflation of tree C.I. values (*Kitching et al., 1998*). The final matrix consists of 52 binary and 28 multi-state characters ten of which were treated as ordered if they were judged to form a morphocline (*Slowinski, 1993*).

Branches were set to collapse and form polytomies if the maximum branch length was zero. This is the default setting for PAUP* and preferable to collapsing minimum branch lengths of zero for this small dataset as the latter method can be too strict for small datasets, eliminating possible topologies (*Swofford & Begle, 1993*; *Coddington & Scharff, 1994*). Nonetheless, a test run with the 'minbrlens' setting was conducted, but obtained the same results as 'maxbrlens', as there is good support for all recovered branches. The matrix was analyzed using the Branch and Bound ('bandb') search option and the resultant trees were rooted with the outgroup *Postosuchus kirkpatricki* ('outroot=para').

A Permutation Tail Probability (PTP) test (*Faith, 1991*; *Faith & Cranston, 1991*) was conducted to test whether the data contain a signal that differs significantly from random data. The result of P=0.01 is demonstrative that the constructed dataset for this study (28 taxa, 83 characters) is significantly more structured than a random dataset (*Faith & Cranston, 1991*; *Hillis & Huelsenbeck, 1992*).

### Results

The initial run of 27 in-group taxa and 83 characters (80 parsimony informative), with the settings given above, yielded 30 most parsimonious trees (MPTs) with a length of 203 steps; a reported Consistency Index (C.I.) of 0.5567, Homoplasy Index (H.I.) of 0.4433, a Retention Index (R.I.) of 0.7345, and a Rescaled Consistency Index (R.C.) of 0.4089. The strict consensus of these trees is provided in Fig. 6A and features a large polytomy at the base of the tree. An Adams consensus (*Adams, 1972*) of the 30 MPTs (Fig. 6B) recovers *Aetobarbakinoides brasiliensis* at the base of this large polytomy, and examination of the 30 MPTs demonstrates that this taxon occurs in 10 possible positions throughout the

strict consensus tree including as the sister taxon to *Revueltosaurus callenderi,* the sister taxon to all aetosaurs, the sister taxon to the Desmatosuchinae, and the sister taxon to the Typothoracinae. A 50% Majority Rule consensus tree (Fig. 6C) places *Aetobarbakinoides* in a polytomy with *Stagonolepis olenkae* and Desmatosuchinae in 70% of the recovered trees. *Coahomasuchus kahleorum* is recovered in three positions in the strict consensus, as the sister taxon to *Aetosaurus ferratus*, the sister taxon to Typothoracinae, and as the sister taxon to *Aetosaurus ferratus* + Typothoracinae.

A reduced consensus tree (Fig. 6D) was generated by pruning *Aetobarbakinoides brasilensis.* Thus, this final matrix has 27 taxa and 83 characters (80 are parsimony informative). The reduced consensus tree has a length of 201 steps, a C.I. of 0.5622, H.I. of 0.4378, a R.I. of 0.7373, and a R.C. of 0.4145.

The reduced consensus (Figs. 6D and 7) features a nearly resolved topology with the exception of a clade with the unresolved polytomy that includes *Coahomasuchus kahleorum, Aetosaurus ferratus*, and Typothoracinae. Bremer support values were calculated for each node utilizing PAUP* by running repeated heuristic searches keeping trees one step longer in each iteration and noting which nodes collapse in strict consensus trees until no nodes remain. No nodes have a support value higher than four and many clades collapse after a single additional step (Fig. 7).

Bootstrap values were calculated using 600 replicates. Because of computational constraints I was unable to calculate bootstrap values using a higher number of replicates. Although using more replicates provides a better representation of confidence values, replicate numbers as low as 100, will provide a "rough but useful estimate" (*Efron, Halloran & Holmes, 1996*: 13432). Bootstrap values for this analysis are provided for all nodes in Fig. 7. Bootstrap values higher than 70%, the minimum meaningful value according to *Hillis & Bull (1993)*, are noted in black, values less than 70% are provided in red, with values lower than 50% interpreted as having very low confidence. Synapomorphy lists for all nodes and definitions of clade names are provided in Appendix C.

*Aetosauroides scagliai* was recovered at the base of the tree as a non-stagonolepidid aetosaurian, similar to the most recent analyses (*Desojo, Ezcurra & Kischlat, 2012*; *Heckert et al., 2015*; *Roberto-da-Silva et al., 2014*). Stagonolepididae (*Heckert & Lucas, 2000*) comprises two major clades, Aetosaurinae (*Heckert & Lucas, 2000*) and Desmatosuchia (clade nov.; Appendix C). The former includes Paratypothoracini (*Parker, 2007*) as the sister taxon to a clade consisting of *Typothorax coccinarum* and *Redondasuchus rineharti.* Paratypothoracini includes *Rioarribasuchus* (=*Heliocanthus*) *chamanensis*, SMNS 19003 (*Paratypothorax* sp. of *Sulej (2010)* & *Desojo & Schoch (2014)*), *Tecovasuchus chatterjeei, Paratypothorax andressorum*, and *Paratypothorax* sp. (North American *Paratypothorax* specimens). This clade is well supported by six unambiguous synapomorphies (Appendix C), as well as a high decay index (+4) and bootstrap value (95%).

The sister taxon to that clade ((*Typothorax* + *Redondasuchus*) + Paratypothoracini) is the recently described *Apachesuchus heckerti* (*Spielmann & Lucas, 2012*), which is known from only a handful of osteoderms, and is situated here based mainly on the presence of the synapomorphy that supports the clade, width/length ratio of widest paramedian osteoderms 3.5 or higher (character 64-2).

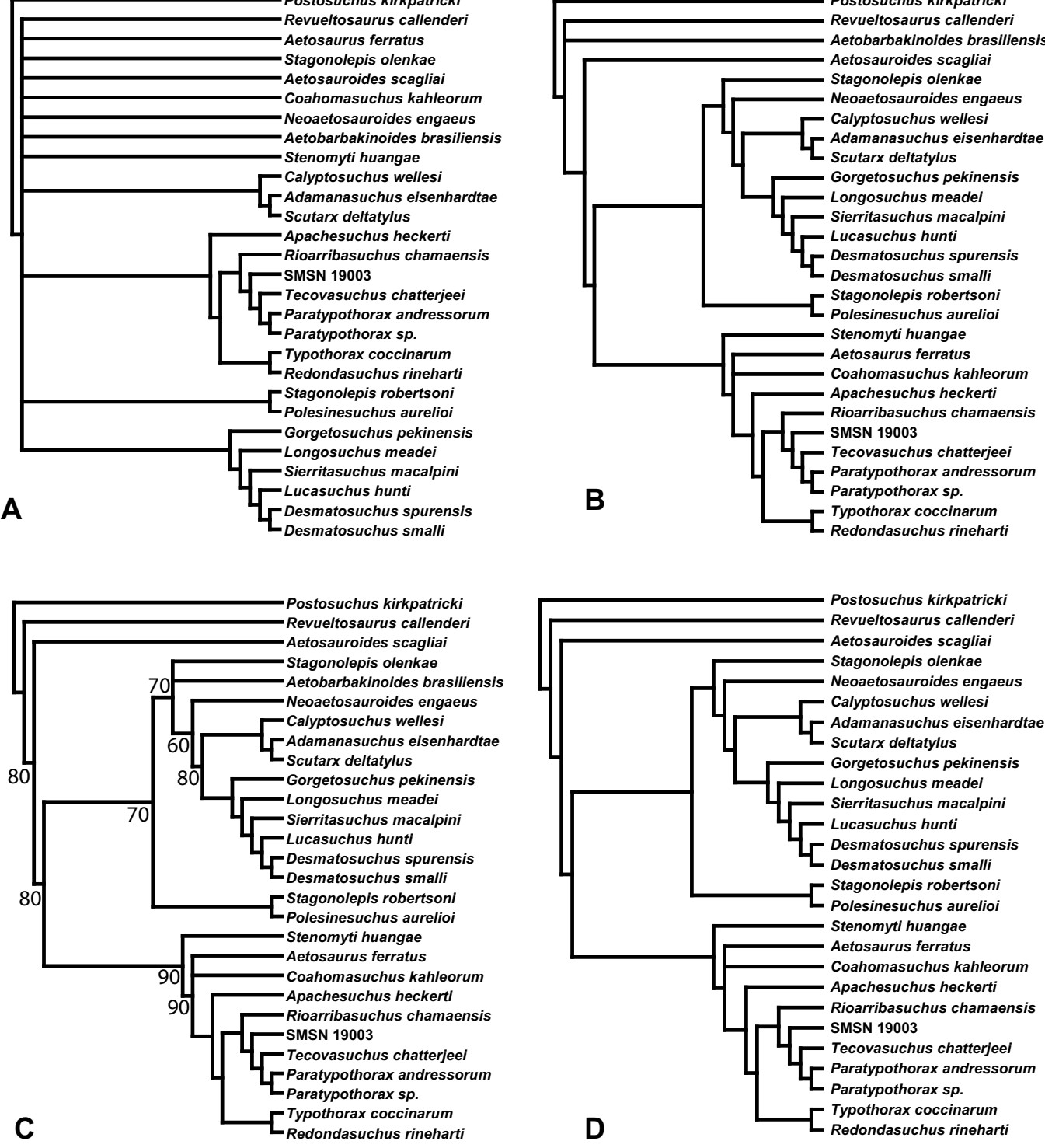

**Figure 6 Phylogenetic trees recovered from the initial run of the main dataset.** (A) Strict component consensus of 30 MPTs; (B) Adams consensus of 30 MPTs; (C) 50% Majority Rule consensus of 30 MPTs. Only values under 100% are shown; (D) Maximum agreement subtree after *a priori* pruning of *Aetobarbakinoides brasiliensis*.

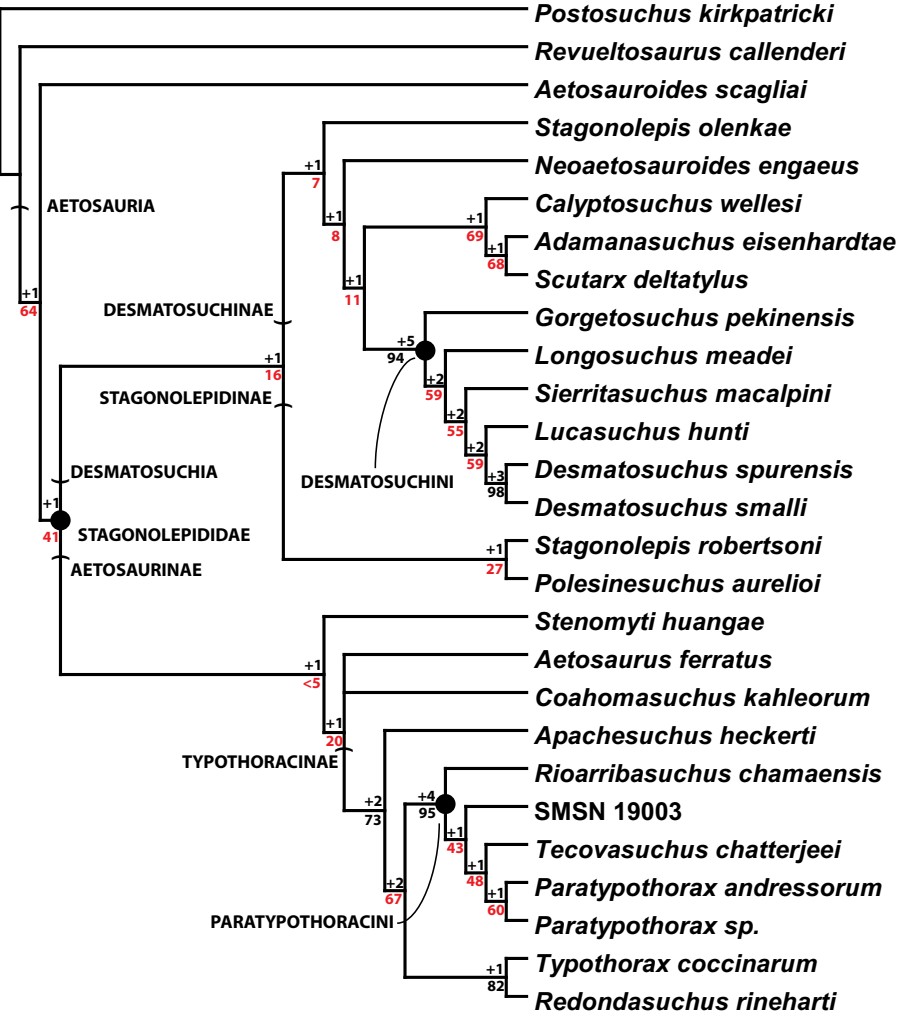

**Figure 7 The reduced strict consensus of 3 MPTs used for this study with *Aetobarbakinoides brasiliensis* removed, with all named clades.** Decay indices and bootstrap values are shown for all nodes, with bootstrap values under 70% (the confidence threshold of *Hillis & Bull (1993)*) shown in red.

In this analysis Typothoracinae, as defined by *Parker (2007)*, would be equivalent to Aetosaurinae, so Typothoracinae is redefined here with an additional specifier (*Aetosaurus ferratus*, see Appendix C). Under this new definition Typothoracinae presently consists of *Apachesuchus heckerti*, Paratypothoracini, and *Typothorax coccinarum + Redondasuchus rineharti*. This clade is well supported by bootstrap values and decay indices (Fig. 7).

As previously mentioned *Aetosaurus ferratus* and *Coahomasuchus kahleorum* form a polytomy with Typothoracinae (Fig. 7). This close relationship is novel, but not entirely unprecedented as these taxa were recovered as adjacent terminal taxa by *Heckert et al. (2015)* & *Roberto-da-Silva et al. (2014)*. Nonetheless, because of the polytomy support for this clade is not robust and these taxa may form other relationships in future analyses. *Stenomyti huangae* (*Small & Martz, 2013*) is recovered at the base of Aetosaurinae, but this position is also very weakly supported and at present there can be little confidence in this position.

Desmatosuchia consists of two clades, Stagonolepidinae (*Heckert & Lucas, 2000*) and Desmatosuchinae (*Heckert & Lucas, 2000*). Stagonolepidinae consists of *Stagonolepis robertsoni* (by definition) and the newly described *Polesinesuchus aurelioi* (*Roberto-da-Silva et al., 2014*), however, this relationship is not very well supported with a decay index of +1 and a bootstrap value of 27% (Fig. 7).

At the base of Desmatosuchinae lie *Stagonolepis olenkae* (*Sulej, 2010*) and *Neoaetosauroides engaeus* (Fig. 7). *Neoaetosauroides* was previously recovered outside of Desmatosuchinae by *Parker (2007)* & *Desojo, Ezcurra & Kischlat (2012)*, but within by *Heckert & Lucas (1999)* & *Heckert & Lucas (2000)*. Regardless these positions are not well supported with both branches having decay indices of +1 and bootstrap values under 10%.

Nested deeper in Desmatosuchinae is a clade consisting of *Calyptosuchus wellesi*, which is the sister taxon to *Adamanasuchus eisenhardtae* + *Scutarx deltatylus* (Fig. 7). These clades are fairly well supported with decay indices of +1 and bootstrap values in the high 60th percentile nearly reaching the confidence threshold of 70% proposed by *Hillis & Bull (1993)*. This is a novel position for these taxa as *Adamanasuchus eisenhardtae* and *Calyptosuchus wellesi* had been recovered outside of Desmatosuchinae in previous studies (e.g., *Parker, 2007*; *Desojo, Ezcurra & Kischlat, 2012*). The presence of these five taxa within Desmatosuchinae is poorly supported with nodes having decay indices of only +1 and bootstrap values of less than 50% (Fig. 7). Thus, this part of the tree may also prove to be highly labile in future analyses.

The subsequent nested clade within Desmatosuchinae; however, is highly supported by 13 unambiguous synapomorphies, a decay index of +5, and a bootstrap value of 94%. I name this clade Desmatosuchini and define it in Appendix C. In this study, Desmatosuchini is well-resolved and includes *Gorgetosuchus pekinensis*, *Longosuchus meadei*, *Sierritasuchus macalpini*, *Lucasuchus hunti*, and *Desmatosuchus*. This new clade has the same constituent taxa as Desmatosuchinae *sensu Parker (2007)*.

## DISCUSSION

### Comparisons to previous analyses

#### Constituency and Status of Major Clades of Aetosauria

Four major clades have been defined within Aetosauria: Stagonolepididae, Aetosaurinae, Stagonolepininae (emended to Stagonolepidinae by *Sereno (2005)*), and Desmatosuchinae (*Heckert & Lucas, 1999*; *Heckert & Lucas, 2000*). A fifth, Typothoracinae, was added by *Parker (2007)*.

Historically the terms Stagonolepididae and Aetosauria have been used interchangeably for family-group names under the Linnaean taxonomic system (see discussion in *Walker (1961)*), but were first defined cladistically by *Heckert & Lucas (2000)*, the former as stem-based and the latter as node based, although in that analysis they contained the same taxa. *Parker (2007)* also recovered these clades at a shared node, but cautioned that the definition provided by *Heckert & Lucas (2000)* was based on *Aetosaurus* occupying the base of the tree and left open the possibility for non-stagonolepidid aetosaurs, which

would alter the historic usage of the name. Rescoring of character states in *Aetosauroides* moved it to the base of Aetosauria as a non-stagonolepidid aetosaur (*Desojo, Ezcurra & Kischlat, 2012*), a position recovered in all subsequent analyses including the present study (*Roberto-da-Silva et al., 2014*; *Heckert et al., 2015*).

In the original defining analysis of *Heckert & Lucas (1999)*, Aetosaurinae included only *Aetosaurus*; however, *Parker (2007)* & *Parker, Stocker & Irmis (2008)* recovered Aetosaurinae as a greatly expanded clade that included all non-Desmatosuchines; however, this clade was generally unsupported and its constituents not accepted by all workers (e.g., *Schoch, 2007*). Moreover, subsequent analyses (*Desojo, Ezcurra & Kischlat, 2012*; *Heckert et al., 2015*) do not recover Aetosaurinae as a more inclusive clade with *Aetosaurus ferratus* the only constituent by original definition. In these analyses, the remnant of the "Aetosaurines" (*sensu Parker, 2007*) are poorly resolved along the spine of Stagonolepididae.

The present study recovers a different result (Fig. 7) with *Aetosaurus ferratus*, *Coahomasuchus kahleorum*, and *Stenomyti huangae*, which was originally referred to the genus *Aetosaurus* (*Small & Martz, 2013*), situated near the base of Aetosaurinae, which also includes the Typothoracinae. This still differs from Aetosaurinae as recovered by *Parker (2007)*, which also included *Stagonolepis robertsoni, Aetosauroides scagliai, Neoaetosauroides engaeus*, and *Calyptosuchus wellesi*, all of which are now recovered as more closely related to *Desmatosuchus* (Fig. 7). However, constraining the analysis to recover all of these taxa in a monophyletic Aetosaurinae (*sensu Parker, 2007*) now requires 11 additional steps.

As defined by *Heckert & Lucas (2000)* Stagonolepidinae consisted of *Stagonolepis robertsoni* and *Coahomasuchus kahleorum*. *Parker (2007)* recovered Stagonolepidinae at the same node as Aetosaurinae and chose to use the latter name for that clade. Subsequently the name Stagonolepidinae has fallen out of use in recent analyses although it would have pertained solely to *Stagonolepis robertsoni* in other recovered topologies (*Desojo, Ezcurra & Kischlat, 2012*; *Heckert et al., 2015*). However, in the present study Stagonolepidinae is distinct from Aetosaurinae as originally conceived and consists of *Stagonolepis robertsoni* and *Polesinesuchus aurelioi* (Fig. 7).

Desmatosuchinae was first recovered as a clade by *Heckert & Lucas (1999)* & *Heckert & Lucas (2000)* where it was comprised of *Desmatosuchus, Typothorax, Paratypothorax*, and *Longosuchus*; however, aspects of the published tree were affected by typographical and scoring errors, as well as reductive coding methods by *Harris, Gower & Wilkinson (2003)*, who provided a revised version of the *Heckert & Lucas (1999)* matrix. The cladogram in *Harris, Gower & Wilkinson (2003)* based solely on the revised *Heckert & Lucas (1999)* matrix recovered Desmatosuchinae as consisting of *Desmatosuchus, Longosuchus, Lucasuchus*, and *Acaenasuchus*, all of which have remained constituent taxa in all subsequent analyses (*Parker, 2007*; *Parker, Stocker & Irmis, 2008*; *Desojo, Ezcurra & Kischlat, 2012*; *Heckert et al., 2015*; this study), although this present study did not include *Acaenasuchus* as an Operational Taxonomic Unit (see explanation above).

The present study differs from all others in recovering several taxa within Desmatosuchinae for the first time, including *Stagonolepis olenkae, Neoaetosauroides engaeus, Adamanasuchus eisenhardtae, Scutarx deltatylus*, and *Calyptosuchus wellesi* (Fig. 7). Nevertheless, support for these included taxa is weak, and it is probable that in

future analyses they may continue to migrate between the bases of Aetosaurinae and Desmatosuchia. A new robust clade, Desmatosuchini, is erected for the taxa originally within Desmatosuchinae (*sensu stricto*) as originally recovered by *Harris, Gower & Wilkinson (2003)* & *Parker (2007)*.

Typothoracinae was first recovered and defined by *Parker (2007)* and is comprised of taxa more closely related to *Typothorax* and *Paratypothorax* than to *Aetosaurus*, *Stagonolepis*, or *Desmatosuchus*. This clade was well-supported by *Parker (2007)* and has been recovered in all subsequent analyses including the present analysis (Fig. 7).

Desmatosuchinae and Aetosaurinae were recovered as sister taxa, with Typothoracinae nested within Aetosaurinae (*Parker, 2007*). *Desojo, Ezcurra & Kischlat (2012)* & *Heckert et al. (2015)* did not recover a similar topology after rescoring and adding taxa to the *Parker (2007)* matrix. Instead they presented trees with Desmatosuchinae and Typothoracinae as sister taxa. The present analysis recovers Typothoracinae within Aetosaurinae and a distinct Desmatosuchinae (Fig. 7).

In sum, the results of five most recent phylogenetic analyses demonstrate that Typothoracinae and Desmatosuchini are robust clades within Aetosauria, well-supported and stable when taxa are added and scorings are changed. Recovery of an inclusive Aetosaurinae is not consistent across studies, with weak support values for non-desmatosuchine and typothoracine taxa causing the constituent taxa to be shuffled around the base of the tree in most studies. The significance of and a possible reason for this are addressed below.

### The Monophyly of Stagonolepis

It has been recognized that aetosaurian material, especially osteoderms, recovered from southwestern North America (Chinle Formation, Dockum Group) is very similar in overall anatomy to that of *Stagonolepis robertsoni*. In fact, the first person to directly compare these materials was convinced of their congeneric status (Charles Lewis Camp, unpublished notes, 1935, in the UCMP archives). The North American material was eventually named *Calyptosuchus wellesi* by *Long & Ballew (1985)*; however, soon afterwards that species was reassigned to the genus *Stagonolepis* (*Murry & Long, 1989*; *Long & Murry, 1995*).

This potential relationship was first discussed in a numerical phylogenetic framework by *Heckert & Lucas (1999*:62*)* who noted that *Calyptosuchus wellesi* and *Stagonolepis robertsoni* "score almost identically throughout the matrix," and therefore they removed *Calyptosuchus wellesi* prior to their final run. For the same reasons they removed *Aetosauroides scagliai*, considering it also to represent *Stagonolepis robertsoni* and several anatomical descriptions were published detailing these proposed synonymies (*Lucas & Heckert, 2001*; *Heckert & Lucas, 2002*). However, investigation of the original matrix by *Harris, Gower & Wilkinson (2003)* determined that because these three taxa were not scored identically, *Calyptosuchus wellesi* and *Aetosauroides scagliai* could not be removed without affecting the final analysis. A reanalysis did not recover a "*Stagonolepis*" clade with *Calyptosuchus wellesi* and *Stagonolepis robertsoni*, but did find a clade with *Stagonolepis robertsoni* and *Aetosauroides scagliai* (*Harris, Gower & Wilkinson, 2003*:fig. 9).

The strict consensus tree published by *Parker (2007*:Fig. 13*)* offered no resolution to this problem, recovering all three taxa in an unresolved polytomy with *Aetosaurus ferratus*. However, *Desojo (2005)* argued against the synonymy of *Aetosauroides* and *Stagonolepis* and in a recent redescription of *Aetosauroides scagliai* demonstrated key differences in the skull and postcranial skeleton that preclude an assignment of that material to *Stagonolepis robertsoni* (*Desojo & Ezcurra, 2011*). More recent phylogenetic analyses featuring a rescoring of *Aetosauroides scagliai* do not recover the three '*Stagonolepis*-like' species as a discrete clade (*Desojo, Ezcurra & Kischlat, 2012*; *Heckert et al., 2015*). The present study rescores *Calyptosuchus wellesi* based on material from the *Placerias* Quarry (*Parker, 2014*) and also does not recover *Stagonolepis*, *Calyptosuchus*, and *Aetosauroides* as a discrete clade. Constraining the present analysis to recover them in an exclusive clade requires 10 additional steps. Thus, anatomical comparisons and several phylogenetic analyses strongly support the separation of these three taxa and the genera *Calyptosuchus* and *Aetosauroides* should no longer be considered junior synonyms of *Stagonolepis* (*Parker, 2008a*; *Desojo & Ezcurra, 2011*).

Numerous well-preserved cranial bones from Krasiejów Poland were described as a new species of *Stagonolepis*, *Stagonolepis olenkae* (*Sulej, 2010*). Postcranial bones and osteoderms were also recovered from the same quarry (*Dzik, 2001*; *Dzik & Sulez, 2007*) and were assigned to *Stagonolepis robertsoni* by *Lucas, Spielmann & Hunt (2007)*. In a traditional (i.e., non-cladistic) analysis *Stagonolepis olenkae* was considered to be an early member of an anagenetic '*Stagonolepis-Aetosaurus*' lineage (*Sulej, 2010*). Differences between *Stagonolepis olenkae* and *Stagonolepis robertsoni* appear to all be in the skull and include contrasting dimensions of various cranial bones, the presence of a massive ridge on the anterior end of the palatine in *Stagonolepis olenkae*, the presence of a lateral ridge on the maxilla of *Stagonolepis robertsoni*, and most notably a reduced number of dentary teeth and the presence of large tubercles on the parietals of *Stagonolepis olenkae* (*Sulej, 2010*). These were considered to possibly represent individual variation by *Antczak (2015)*, who suggested that the Krasiejów material is probably referable to *Stagonolepis robertsoni*. In the phylogenetic analysis presented here these two taxa are scored differently for five characters, four are cranial and the fifth is that the humeral head is more expanded in *Stagonolepis olenkae*. In the recovered tree *Stagonolepis robertsoni* + *Polesinesuchus aurelioi* is the sister taxon to *Stagonolepis olenkae* + Desmatosuchinae. A topological constraint to force the two purported species of *Stagonolepis* to form an exclusive clade requires only an additional two steps. Therefore, even though both purported species were not recovered as a clade, I do not suggest erecting a new generic name to receive *Stagonolepis olenkae*. Differences between the taxa are too few and potentially explained by the much larger size of *Stagonolepis olenkae*, although *Sulej (2010)* explicitly argued against this possibility. A full description of the postcranial material and osteoderms will hopefully provide further evidence for or against the potential generic synonymy of these two taxa although a preliminary analysis proposes synonymy (*Antczak, 2015*).

### The Phylogenetic Position of Aetosaurus ferratus

The earliest exhaustive treatment of the Aetosauria (*Walker, 1961*) considered *Aetosaurus ferratus* as the 'basal' aetosaurian, a position supported by the first phylogenetic analyses

of the Aetosauria (*Parrish, 1994*; *Heckert, Hunt & Lucas, 1996*; *Heckert & Lucas, 1999*). Indeed, an early study constrained *Aetosaurus ferratus* to this position by utilizing it as the sole outgroup for the entire analysis (*Heckert, Hunt & Lucas, 1996*). Nonetheless that study considered other aetosaurs to be more 'advanced' than *Aetosaurus* based on characters of the teeth, especially the presence of bulbous rather than recurved teeth and an edentulous anterior portion of the dentary. Those characters and scorings for *Aetosaurus* were taken directly from *Parrish (1994)*, and used again by *Heckert & Lucas (1999)* to diagnose *Aetosaurus*. *Parker (2007)* followed *Walker (1961:164)* in considering the teeth of *Aetosaurus* bulbous, rather than mediolaterally flattened and recurved, with the anterior portion of the dentary edentulous. In the accompanying analysis, *Aetosaurus ferratus* was recovered more deeply nested within Stagonolepididae (*Parker, 2007*), the first time it had not been recovered at the base of Aetosauria in a phylogenetic analysis, (*Parrish, 1994*; *Heckert, Hunt & Lucas, 1996*; *Heckert & Lucas, 1999*). This alternate placement prompted detailed discussion by *Schoch (2007)* who acknowledged that the teeth were as *Walker (1961)* had described, but argued that the more nested placement of *Aetosaurus* was somewhat ambiguous as other character states found in *Aetosaurus ferratus* supported a position closer to the base of Aetosauria.

In subsequent analyses (*Desojo, Ezcurra & Kischlat, 2012*; *Heckert et al., 2015*), *Aetosaurus* has been recovered closer to the base of Aetosauria in part mainly because of a change in character polarities based on the scoring of *Aetosauroides scagliai* as having a maxilla that is excluded from the margin of the external naris (*Desojo & Ezcurra, 2011*); a change that pulled both *Aetosauroides* and *Aetosaurus* towards the root of the tree. In the present analysis *Aetosaurus* is recovered in a polytomy with *Coahomasuchus* and Typothoracisinae, and two taxa are still fairly close to the base of Aetosauria (Fig. 6), but constraining the clade of *Aetosaurus* plus *Coahomasuchus* to the base of Aetosauria requires an additional six steps.

### Low Support Values in Data Partitions

Overall, the tree of *Heckert et al. (2015)* is the most similar of all past studies to the novel one presented here, suggesting that the incorrect scorings that affected the earliest analyses (*Parrish, 1994*; *Heckert, Hunt & Lucas, 1996*; *Heckert & Lucas, 1999*) still played a major role in the recovered topology of *Parker (2007)*. Some of these errors were directly inherited from the previous studies (*Parrish, 1994*; *Heckert, Hunt & Lucas, 1996*), but others resulted from a general lack of good specimens and a necessary reliance on outdated literature to score characters as redescriptions of key taxa such as *Aetosaurus ferratus*, *Aetosauroides scagliai*, *Neoaetosauroides engaeus*, and *Desmatosuchus spurensis* had not yet been published (*Desojo & Báez, 2005*, *Desojo & Báez, 2007*; *Schoch, 2007*; *Parker, 2008b*; *Desojo & Ezcurra, 2011*). Still, this early work should be recognized for pioneering phylogenetic studies of aetosaurians, especially the study of *Heckert, Hunt & Lucas (1996)*, which introduced many key characters still used in current analyses. However, this also demonstrates the importance of discovering and utilizing new specimens of existing taxa (e.g., MNA V9300, PEFO 38001/YPM 58121; NMMNH P-56299; TMM 31100-437), as well as crucial reinvestigations of original

type materials (e.g., *Desojo & Báez, 2005*; *Desojo & Báez, 2007*; *Schoch, 2007*), in phylogenetic work.

I find the results of the new study presented here to be generally disappointing because of the lack of support for the base of the tree, essentially all nodes outside of Typothoracinae and Desmatosuchini. This problem also plagued the previous study by *Parker (2007)* and was apparent in the way topologies shifted significantly in new studies when characters were rescored and new taxa added (*Desojo, Ezcurra & Kischlat, 2012*; *Heckert et al., 2015*). The present work sought to increase character support by creating as many new characters as possible, particularly those from skeletal elements outside of the dorsal carapace. Inclusion of endoskeletal (non-armor) characters was suggested as a way to provide tree stability (*Desojo, Ezcurra & Kischlat, 2012*; *Heckert et al., 2015*).

*Parker (2007)* scored 35 parsimony informative characters with 23 (66%) of these characters from the osteoderms. This new study has expanded the dataset to 80 parsimony informative characters, an increase of over 100%, with only 31 of these characters scoring osteoderm characters (39%). Thus, it was expected to see an increase in stability in the overall tree metrics utilizing a dataset with better skeletal region sampling, but unfortunately this was not realized in the final results.

One of the possible reasons for these low support values is that the non-osteoderm characters of aetosaurians appear to generally have higher levels of homoplasy. For example, the 35 parsimony informative cranial characters have an average C.I. value of 0.596. This value was computed by simply adding up the C.I. scores for each character and dividing the resulting number by the number of characters, thus this is a calculation of a 'raw' C.I. average and does not equate the final reported C.I. number for the MPTs as determined by PAUP*. Vertebral characters score much higher with an average C.I. value of 0.767. However, the paramedian osteoderm characters have an average value of 0.697, whereas the lateral osteoderms have an average value of 0.833 demonstrating the value of the osteoderm characters. Overall the non-osteoderm characters have an average C.I. value of 0.606, compared to an average value of 0.746 for the osteoderm characters. What this signifies is that the non-osteoderm characters included in the study are more apt to change across the tree than the osteoderm characters, which signifies a higher degree of homoplasy in non-osteoderm characters or that non-comparable maturity at time of death among specimens plays a greater role than expected.

Overall, sampling of non-osteoderm characters remains poor, with the majority of characters taken from the cranial region. Only four characters sample the appendicular skeleton, and the limbs and girdles represent a potential area for character expansion. Unfortunately, many aetosaur taxa do not have limb and girdle material referred to them, and, in some cases, these materials are present but covered by an articulated carapace and not accessible for study without non-invasive (e.g., CT) scanning. Where limb and girdle elements are known (e.g., femora, scapulae, ilia) many of the characters appear to be conserved across taxa. Still, with increasing sample sizes and better comparisons, more informative characters can probably be derived from this dataset in future analyses.

## Dataset Partitioning

An interesting question brought up during the construction of this dataset is: what if aetosaurians did not possess an extensive armor carapace? What if all of the characters and character states used in phylogenetic analyses of the Aetosauria were derived from the skull, axial, and appendicular portions of the skeleton as is the case for the majority of vertebrate animals? In sum, what would the phylogeny of aetosaurians look like without utilizing characters of the osteoderms?

One particular aspect of phylogenetic data set analysis is the discussion of data partitioning, which entails the separation of a data set of phylogenetic characters into discrete parts based on types of characters (e.g., molecular sequences vs. morphological; *Kluge, 1989*), or modular (e.g., skull vs. postcranium; *Mounce, 2013*). In most published cases, the debate over data partitioning in phylogenetic analyses revolves around the advantages or disadvantages of combining molecular sequence and morphological data into a single data set (e.g., *Bull et al., 1993*). Considerable discussion is available regarding partitioning of strictly morphological data into discrete character sets based mainly on anatomical subregions (*Rowe, 1988*; *Gauthier, Kluge & Rowe, 1988*; *Donoghue et al., 1989*; *Rae, 1999*; *Clarke & Middleton, 2008*; *Mounce, 2013*), but none pertains to the special case of osteoderms versus endoskeletal features.

Aetosaurians provide an excellent example of a group where historically the taxonomy is based almost entirely on characters from a single non-cranial anatomical subregion. A major assumption of aetosaur workers is that not only is osteoderm anatomy taxonomically informative, but that it is also phylogenetically significant, providing an accurate signal of evolutionary relationships within the group (*Parker, 2007*). Dataset partitioning provides a test of which characters, in this case integumentary versus non-integumentary, are providing the main phylogenetic signal for this group and allows for in-depth examination of possible underlying factors regarding the poorly resolved phylogenetic relationships recovered in past studies (*Harris, Gower & Wilkinson, 2003*).

### Why partition?

Osteoderms represent a mineralized component of the dermis in tetrapods (*Romer, 1956*; *Hill, 2005*). As such they are hypothetically an autonomous (i.e., they are not found in all vertebrates) unit (module) of the skeletal and circulatory systems. This independence is further supported by the finding that onset of osteoderm development is delayed, by as much as a year, in comparison with the rest of the skeleton in *Alligator* with the result that they are absent in very young animals (*Vickaryous & Hall, 2008*). This independence also suggests that the osteoderms, with specific proposed functions (e.g., defense, heat transfer, species recognition; *Seidel, 1979*; *Main et al., 2005*), may be under different evolutionary selection pressures than other parts of the body such as the head, which is mainly focused on resource acquisition, or the limbs, which are mainly focused on locomotion and/or environmental manipulation. Thus, they can be considered a distinct module of the skeleton. This begs the question of how does the non-integumentary portion of the aetosaur skeleton compare to other taxonomic groups, but more importantly how does it compare within Aetosauria proper?

## Methods

Morphobank (*O'Leary & Kaufman, 2012*) was used to divide the main dataset into smaller partitions based on cranial characters, osteoderm characters, and the full set of non-osteoderm characters. The cranial dataset consists of characters 1–35, the osteoderm dataset consists of characters 52–83, and the full endoskeletal set consists of characters 1–51. All analyses for this study were run using PAUP* version 4.0.b10 (*Swofford, 2003*). All characters were weighted equally and the most parsimonious trees (MPTs) were subject to an exact solution search using the branch and bound implementation under the program default settings. Bootstrap support values for each dataset were calculated from 1000 replicates with only scores above 50% being recorded as informative, although only values above 70% probably represent well-supported clades (*Hillis & Bull, 1993*). Distribution of character states was analyzed in Mesquite 2.75 (*Maddison & Maddison, 2011*).

The reduced consensus tree provided earlier utilized the full data set for this entire project (26 in-group taxa and 83 characters) and establishes the baseline relationships for this study. For this portion of the study, runs used the subsets outlined above. Because several taxa are only known from osteoderms (e.g., *Apachesuchus heckerti*, *Redondasuchus rineharti*, *Rioarribasuchus chamaensis*) it was necessary to remove taxa where no characters could be scored for one of the partitions, as inclusion of taxa with no scored characters causes the algorithm to generate all possible trees, which increases exponentially given the total number of taxa with the end result of a completely unresolved consensus tree. Therefore, all taxa lacking skull material were also removed from the matrices so that the final trees could be directly compared. Taxa were also removed if taxonomic equivalence could be demonstrated, utilizing the Safe Taxonomic Reduction method of *Wilkinson (1995a)* to reduce the number of MPTs and increase resolution. For the cranial set this included *Tecovasuchus chatterjeei*, which can only be scored for two characters, and for *Desmatosuchus spurensis*, which for this partition is identically coded to *Desmatosuchus smalli*. *Desmatosuchus smalli* was retained for the analysis as its overall scoring contains fewer missing data (98% complete). For the osteoderm-only dataset this included *Stagonolepis olenkae*, which is scored identical to *Stagonolepis robertsoni*, *Desmatosuchus spurensis* which is scored the same as *Desmatosuchus smalli*, and *Redondasuchus rineharti*, which is scored the same as *Typothorax coccinarum*. The final partition datasets initially contain 13 taxa. *Postosuchus kirkpatricki* and *Revueltosaurus callenderi* are utilized as the outgroup, and the in-group taxa consist of *Aetosaurus ferratus*, *Stagonolepis robertsoni*, *Scutarx deltatylus*, *Aetosauroides scagliai*, *Coahomasuchus kahleorum*, *Desmatosuchus smalli*, *Longosuchus meadei*, *Neoaetosauroides engaeus*, *Typothorax coccinarum*, SMNS 19003, and *Stenomyti huangae*.

A 'Simultaneous Analysis' dataset (all 83 characters; *sensu Baker & DeSalle, 1997*), although with the reduced number of taxa (13), was run to see the effects of reduced taxon sampling, which has been hypothesized to reduce phylogenetic accuracy (*Hillis, 1998*), and to establish a baseline topology for comparison with the partitioned datasets. Nonetheless, it should be recognized that partitioning datasets is an analytical tool and should not be expected to represent the final phylogenetic hypothesis. This simultaneous

analysis matrix was subsequently portioned into three datasets, one including only cranial characters, another including only osteoderm characters, and a third including all non-osteoderm characters including the cranial set. All uninformative and constant characters were excluded, further reducing the sizes of the matrices to less than half of the taxa originally scored (13 taxa, 35 characters from the cranial set; 13 taxa, 24 characters for the osteoderm only set, and 13 taxa, 46 characters for the non-osteoderm only set). Because the datasets were reduced, Permutation Tail Probability (PTP) tests were run in PAUP* to test the null hypothesis that the datasets are no better than random and thus phylogenetically uninformative (*Faith & Cranston, 1991*). The cranial and the armor only datasets had PTP scores of 0.01 and the endoskeletal dataset had a score of 0.02 which are less than the required PTP of 0.05 that is considered to be significant, thus falsifying the null hypothesis. These datasets were run under the branch and bound settings in PAUP* and the results of the partition sets were compared with each other, as well as to the full and reduced taxon datasets containing all 83 characters.

### Results

Reduction of the number of taxa in the full working matrix from 28 to 13 taxa produced 14 parsimony-uninformative characters (including four constant characters) out of the original set of 83. The uninformative characters were excluded *a priori* from the final matrix of 69 characters and 12 in-group taxa. Ten characters were unordered. The initial run (branch and bound) resulted in three most parsimonious trees of 178 steps. The strict consensus of which (C.I. = 0.6067, H.I. = 0.3933, R.I. = 0.5270, R.C. = 0.3198), which is provided in Fig. 8A. This tree is similar to the reduced strict consensus tree recovered in the complete analysis presented earlier except that the base of Aetosauria is unresolved, consisting of *Stenomyti*, *Stagonolepis*, Aetosaurinae and Desmatosuchinae. Nonetheless, taxa recovered in Aetosaurinae and Desmatosuchinae in the full analysis are recovered in those same clades in this reduced analysis. Thus, the reduction of taxa as well as the loss of the 14 constant/uninformative does not significantly change relationships within the tree.

Overall clade support in the base of the reduced matrix tree is not good, with some clades collapsing with only a single additional step. However, Typothoracinae (*Typothorax* + SMNS 19003) collapses after six steps, and Desmatosuchini (*Desmatosuchus* + *Longosuchus*) is particularly well-supported, not collapsing until nine additional steps. Thus, there appears to be no negative effects to clade support as a result of reduced taxon sampling as the nodes are even better supported than in the tree recovered for the complete analysis. Thus, this reduced matrix tree provides a suitable baseline topology to compare to the other partition sets.

A branch and bound search of the reduced matrix utilizing the osteoderm only dataset (12 in-group taxa, 24 informative characters, eight ordered; outgroup constrained) results in a strict consensus tree (Fig. 8B) from three MPTs of 58 steps each (C.I. = 0.8276, R.I. = 0.7727, R.C. = 0.6395). These metrics are high, suggesting that there is reduced homoplasy in this data partition (H.I. = 0.1724). Nonetheless, the recovered tree topology is mostly unresolved and poorly supported. Four clades are recovered; 1) *Desmatosuchus smalli* + *Longosuchus meadei*, which is the sister taxon to all of the other

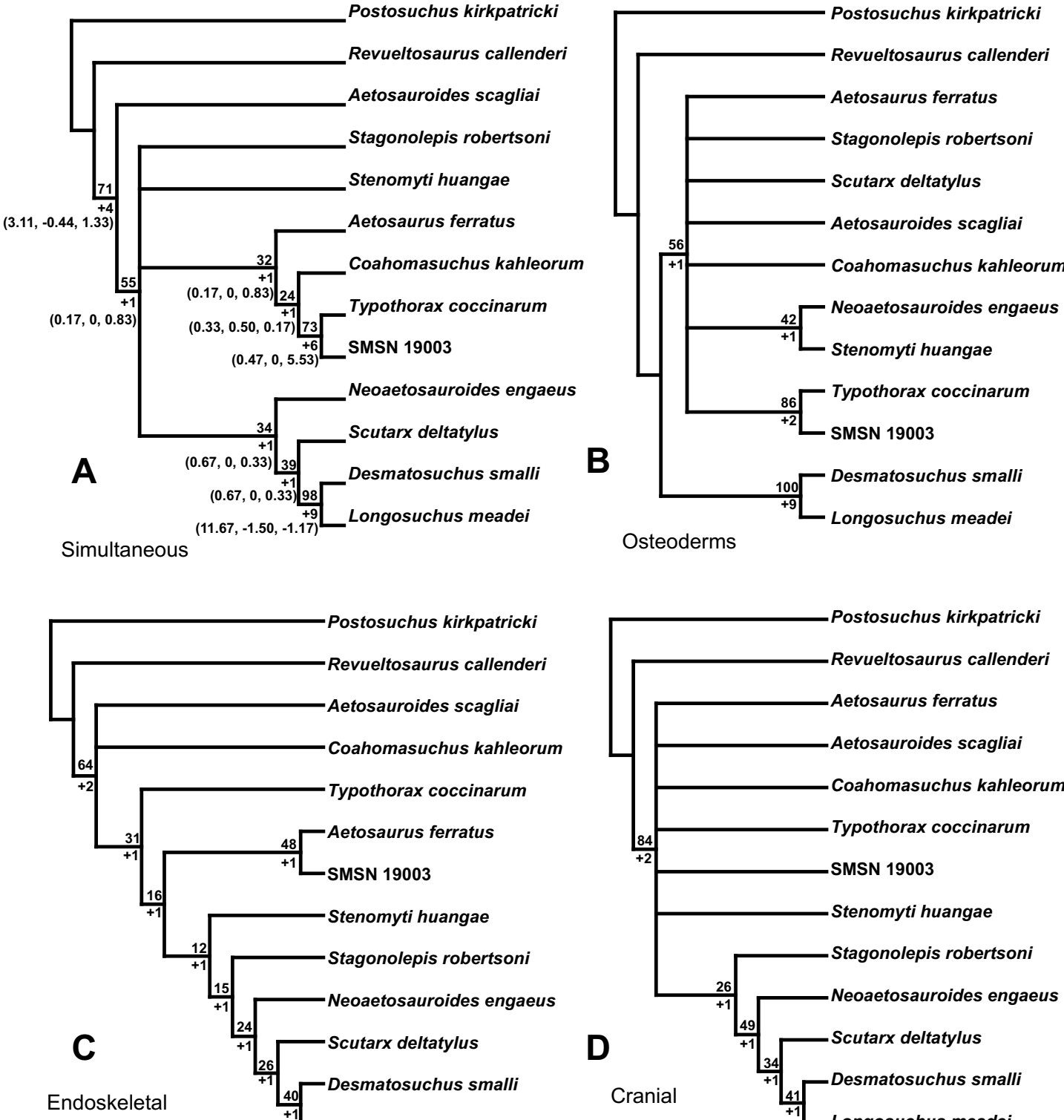

**Figure 8 Phylogenetic trees recovered from partitioning the main dataset.** Decay indices and bootstrap values (1000 replicates) listed for all nodes. (A) Topology of a three MPTs from the simultaneous (13 taxa, 83 characters) dataset. Partitioned Bremer Support values for nodes are given in parentheses. The first value pertains to the cranial only characters, the second from the postcranial characters, and the third from the osteoderm characters; (B) Topology of a three MPTs recovered for the osteoderm dataset; (C) Strict consensus tree from two MPTs from the complete non-osteoderm (endoskeletal) dataset (cranial, axial, appendicular); (D) Strict consensus of 13 MPTs from analysis of the cranial dataset.

aetosaurians; 2) all of the non-desmatosuchine taxa; 3) *Stenomyti huangae* + *Neoaetosauroides engaeus*, and 4) *Typothorax coccinarum* + SMNS 19003 (Fig. 7B). In this partitioned analysis *Stenomyti* + *Neoaetosauroides* is supported by two unambiguous synapomorphies, dorsal eminence of the dorsal paramedian osteoderms is strongly offset medially (66-2), and anterolateral projection of the anterior bar of the dorsal paramedian osteoderms is present and elongate (68-1). *Typothorax coccinarum* + SMNS 19003 are supported by six unambiguous synapomorphies: 1) lateral edge of the dorsal paramedian osteoderms in dorsal view are strongly sigmoidal with a strongly posteromedially oriented posterolateral corner (63-1); 2) width/length ratio of widest paramedian osteoderms (rows 9–11) in dorsal trunk series is greater than 3.5 (64-2); 3) dorsal eminence of cervical lateral osteoderms is a moderate length, dorsoventrally flattened, slightly recurved spine (74-1); 4) mid-dorsal lateral osteoderms with a strongly acute angle of flexion between the dorsal and lateral flanges (79-2); 5) lateral flange of pelvic and anterior caudal lateral osteoderms is roughly triangular in lateral vie with a semicircular ventrolateral border and a hook-like eminence (81-1); and 6) carapace is broad and discoidal in dorsal view (82-2).

*Desmatosuchus* plus *Longosuchus* (Desmatosuchini) is the best supported clade with 14 unambiguous synapomorphies: 1) cervical paramedian osteoderms are longer than wide (57-1); 2) ratio of cervical vertebrae/paramedian osteoderms significantly less than 1:1 (58-1); 3) adjacent paramedian and lateral cervical osteoderms are often fused (59-1); 4) in the paramedian osteoderms dorsal to the cervical and anterior trunk vertebrae, lateral edge articulation with lateral osteoderms is dorsoventrally thickened, angled contact, with deeply incised interdigitation (='tongue and groove') (60-1); 5) dorsal eminence shape in the cervical paramedian osteoderms are a low pyramidal or rounded boss, or elongate keel (61-1); 6) in the dorsal trunk paramedian osteoderms the anterior edge of the lateral osteoderm overlaps the anterior edge of the paramedian osteoderm (62-1); 7) lacks the sharp anteromedial projection of the anterior bar (reversed in *Lucasuchus hunti*) (67-1); 8) the anterior bar of the trunk distal paramedian osteoderms lacks an anterolateral projection (68-2); 9) anterior bar of the dorsal trunk paramedian osteoderms lacks scalloping of the anterior margin on the medial side of the osteoderm (69-1); 10) dorsal eminence in the mid-trunk osteoderms is a conical spike (78-2); 11) approximately 90 degree angle between the dorsal and lateral flanges of the mid-trunk lateral osteoderms (79-1); 12) dorsal trunk lateral osteoderms strongly asymmetrical with the dorsal flange longest (80-1); 13) lateral flange of the pelvic and anterior caudal lateral osteoderms are rectangular and ventral to a well-developed spine (81-2); and 14) overall shape in of the dorsal carapace in dorsal view is moderately spinose (82-1).

Overall support is mixed with the clades *Neoaetosauroides* + *Stenomyti*, and *Typothorax* + SMNS 19003 having Decay Indices of +1 and +2 respectively, but *Desmatosuchus* plus *Longosuchus* is very strongly supported with a Decay Index of +9. Furthermore, *Typothorax* + SMNS 19003 has a bootstrap value of 86% for 1000 replicates, and *Desmatosuchus* plus *Longosuchus* occurs in 100% of the replicates (Fig. 8B).

A branch and bound run of the endoskeletal (non-osteoderm) dataset (12 in-group taxa, 46 informative characters, four ordered) results in two MPTs of 115 steps

(C.I. = 0.5217, H.I. = 0.4783, R.I. = 0.4762, R.C. = 0.2484), the strict consensus of which is shown as Fig. 8C. The tree is nearly completely resolved and the topology more closely resembles the total evidence tree rather than the cranial only tree. *Aetosauroides* is recovered at the base of the tree, and the clade (*Neoaetosauroides* + (*Scutarx* + (*Desmatosuchus* + *Longosuchus*))) is recovered. A significant difference, however, is that SMNS 19003 forms a novel clade with *Aetosaurus* in this partition tree rather than with *Typothorax*. Therefore, the clade Typothoracinae is not supported by this character set. Support for this topology is weak with only clade (Aetosauria) with a bootstrap value higher than 50%. *Aetosaurus* + SMNS 19003 has a Decay Index of +1, a bootstrap value of 48%, and is supported by three unambiguous synapomorphies: 1) ventrolateral margin of the nasal forms part of the dorsal border of the antorbital fossa (10-1); 2) postorbital contacts quadratojugal (15-1); and 3) supratemporal fenestra is greatly reduced in size compared to the orbit (22-2).

A subset of the non-osteoderm dataset, consisting of only cranial characters, was also run using the branch and bound search criteria. This run (12 in-group taxa, 35 informative characters, four ordered) resulted in thirteen MPTs of 82 steps (C.I. = 0.5488, H.I. = 0.4512, R.I. = 0.5542, R. C. = 0.3041) shown as Fig. 8D. This tree is most similar to the non-osteoderm dataset tree, but less resolved. The base of the tree is a large polytomy with *Aetosaurus*, *Aetosauroides*, *Typothorax*, *Coahomasuchus*, SMNS 19003, and *Stenomyti*. *Longosuchus* and *Desmatosuchus* form a clade (Desmatosuchini) with *Scutarx*, *Neoaetosauroides*, and *Stagonolepis robertsoni* as successive sister taxa. Support is no better than in the endoskeletal (non-osteoderm) tree, with all clades a Decay Indices of +1 and a bootstrap values less than 50% (Fig. 8D). As with the endocranial set Typothoracinae is not recovered. However, neither is the clade *Aetosaurus* + SMNS 19003, which was recovered in the endocranial set.

### Dataset Incongruence

A partition homogeneity test was conducted in PAUP* for the 'simultaneous analysis' matrix (excluding uninformative characters following the recommendations of *Lee, 2001*) divided into three partitions (osteoderm, postcranial, and cranial) using the CHARSET command in PAUP*. The test resulted in a p-value score of 0.03 suggesting that some character conflict exists between the partitioned datasets. Incongruence Length Difference (ILD) tests (*Farris et al., 1995*) were run for each partition set, comparing each to the other two sets. The test without the cranial set had a p-value score of 0.70, that without the endoskeletal (non-osteoderm) set had a score of 0.08, and the test excluding the osteoderm set had a p-value of 0.35. These results all show significant incongruence over the 0.05 threshold. These numbers also suggest that although the cranial and osteoderm sets are the most compatible, with low conflict, the osteoderm and postcranial sets and the cranial and postcranial sets have high levels of conflict. Size differences between the partitions (i.e. number of characters) do not influence the ILD test, thus datasets with higher amounts of characters do not 'overwhelm' partitions with lower numbers of characters (*Farris et al., 1995*; *Baker, Yu & DeSalle, 1998*). Therefore, these scores are the result of dataset incongruence.

*Bull et al. (1993)* argued that dataset partitions with high levels of character conflict should not be combined for analyses (the prior agreement approach), whereas others (e.g., *Kluge, 1989*; *Barrett, Donoghue & Sober, 1991*) argue that data should be combined in all cases (the total evidence approach). Still others argued that these debates have been mostly theoretical and it is important to examine the actual data to understand the consequences of these approaches (*Baker & DeSalle, 1997*). The Partition Homogeneity Test (and ILD) measures levels of disagreement between partitions, but does not identify specific nodes where this conflict occurs (*Lambkin et al., 2002*). *Baker & DeSalle (1997)* developed a new measure, Partitioned Bremer Support (PBS) to determine the amount of support individual data partitions contribute to the branch support of the full matrix. Partition datasets that conflict with other datasets at the same node will contribute negatively to the overall branch support. Therefore isolating Bremer Support values for nodes by partition allows for the determination of localized areas of disagreement. The higher the negative PBS number, the greater the support that partition provides for an alternative node that is not present in the combined data tree (*Lambkin, 2004*; *Brower, 2006*). Moreover, strong variance in PBS scores for nodes, demonstrates conflict between partitions for node resolution (*Lambkin, 2004*). Neutral (0) scores indicate that there is within-dataset incongruence and that the particular partition is ambivalent about the node, reducing overall support (*Lambkin et al., 2002*).

The program TreeRot.v3 (*Sorenson & Franzosa, 2007*) was employed to calculate PBS values for the partitioned dataset. This method works back and forth between the TreeRot.v3 program and PAUP*. First the 'simultaneous analysis' matrix is run in PAUP* using the same parameters as the earlier run (12 in-group taxa, 69 informative characters, ten characters ordered, branch and bound search) with the three partitions set-up using the CHARSET command. PAUP* was also used to calculate Bremer Support (BS) values for the entire dataset. The resulting tree file is then entered into TreeRot to generate a PAUP* command file, which includes the Partitioned Bremer Support (PBS) values. Minimum, maximum and averaged values are given for each partition at each node. *Baker & DeSalle (1997)* recommended utilizing the averaged value, but *Lambkin et al. (2002)* argued that averaging masks some of the conflict found at each node. However, for this study I did use the averaged values because it is the averaged values for each partition that sum to match the Bremer Support value listed for each node. The values for this analysis are provided for branches in Fig. 8A. There are three numbers listed, the first is from the cranial character set, the second from the postcranial (vertebrae, girdles, limbs) character set, and the third from the osteoderm character set. Note that the three PBS values equal the total BS value for that branch and as mentioned earlier negative numbers denote negative support (homoplasy) and neutral numbers indicate node ambivalence for that dataset. Also note that these character set (CHARSET) divisions are for the purpose of determining the PBS and do not pertain directly to the partition dataset trees presented in Figs. 8B–D.

The cranial character set supports eight nodes, showing no conflict with the other character sets, although support is low for four of these nodes (below +0.5). The

postcranial character set supports only a single node (*Coahomasuchus* + Typothoracinae), but is mostly neutral except for two nodes where it shows moderate conflict with the other datasets, especially in one node, *Desmatosuchus* + *Longosuchus* (=Desmatosuchini), which has a PBS of −1.50. The osteoderm character set shows positive, but low, for seven out of eight nodes, including good support (+5.53) for Desmatosuchinae. The osteoderm character set shows conflict for Desmatosuchini (−1.17). In sum, character dataset conflict occurs in two nodes, Aetosauria and Desmatosuchini, with all conflict occurring with the postcranial and osteoderm datasets (Fig. 8A) meaning that these partitions better support alternative phylogenies.

### Discussion

Dataset partitioning and partition homogeneity tests (PHT) strongly suggest that the main character suites (i.e. cranial, postcranial, osteoderm) possess some conflicting phylogenetic signals. The PHT suggests that the postcranial dataset conflicts the most with the other datasets, and Partitioned Bremer Support analyses identify the nodes where this conflict exists. This demonstrates that different anatomical modules (e.g., cranium, carapace) may be evolving at different rates because they are under different selective pressures (*Simpson, 1950*).

It had been suggested by previous studies that adding more non-osteoderm character data would stabilize weakly supported and labile relationships outside of the Typothoracinae and Desmatosuchini (*Desojo, 2005*; *Desojo, Ezcurra & Kischlat, 2012*; *Roberto-da-Silva et al., 2014*; *Heckert et al., 2015*), but doubling the size of the matrix and increasing the number of endoskeletal characters to be dominant did not create more support, these inner tree relationships still remain weakly supported, and there is little confidence in the recovered clades. It is presently unclear how stable these nodes will be. Lack of support and accuracy could be caused by the need for more taxon sampling or by large amounts of missing data (*Wiens, 1998b*; *Heath, Hedtke & Hillis, 2008*), but it is also possible that it is caused by incongruence between and within character suites (*Lambkin et al., 2002*). Moreover, missing or inapplicable data has been argued to cause ambiguous character optimizations at nodes (*Ezcurra, Scheyer & Butler, 2014*).

Using the total evidence approach of *Kluge (1989)* and adding more solid character data may overcome dataset noise (*Barrett, Donoghue & Sober, 1991*), and this should be tested in future analyses. Furthermore, the present matrix contains little data from the appendicular skeleton, where the characters appear to be well-conserved, or what differences are apparent cannot be viewed outside of the realm of ontogenetic or sexual variation, but this should be a source of future characters where possible. Increased taxonomic sampling from future discoveries, including the potential discovery of other suchian taxa with lateral armor to serve as improved outgroup taxa, will undoubtedly help improve dataset resolution.

*Bull et al. (1993)* argued that combining heterogeneous datasets can result in an erroneous parsimony estimates and that it is better to keep these data separate to avoid getting a single wrong answer. Data that fail statistical tests for heterogeneity should not be combined and used in analyses that assume the data to be homogeneous, because

character datasets that appear to be independent may in fact be the result of two distinct histories of character change (*Bull et al., 1993*). However, according to those authors *Hillis (1987)* argued that because some character sets may be useful in resolving certain areas of the tree, all data should be combined. If incongruent datasets are combined, any underlying positive signal will be amplified and can often cancel out dataset noise (*Lee, 2009*).

In the tree recovered in the main part of this study (Fig. 7) it is encouraging that the topology 'makes sense,' that is that there is nothing in the topology that would be a major surprise to an aetosaur worker, suggesting that an underlying positive signal is present. For example, *Scutarx deltatylus* and *Calyptosuchus wellesi*, are recovered in the same clade, which is expected as several specimens of *Scutarx* had initially been assigned to *Calyptosuchus* (e.g., *Parker & Irmis, 2005*; *Parker & Martz, 2011*; *Martz et al., 2013*). *Stenomyti huangae* and *Aetosaurus ferratus* are recovered close together just outside of Typothoracinae (Fig. 7) and therefore presents a proposed relationship with little statistical support, yet when originally discovered the material of *Stenomyti* was originally assigned to *Aetosaurus* (*Small, 1998*) and utilizing only anatomical comparisons it would be expected for the two to be recovered close together, again suggesting an underlying positive signal. In contrast, the tree presented by *Parker (2007)* introduced two strong clades (Typothoracinae, Desmatosuchinae), but generally the overall recovered topology did not 'make sense' in regards that, 1) no terminal taxa stemmed from the base of the tree (i.e. there is no 'basal' species-group taxon), and 2) outside of the two clades, all of the other taxa were an unresolved polytomy unsupported by synapomorphies other than a few armor characters.

Nonetheless, caution is warranted when equating 'sense' with accuracy as a specific tree cannot be preferred simply because it meets preconceived notions. For example, at one time it was thought that taxa with a radial ornamentation on the paramedian osteoderms, and similar lateral osteoderm anatomy formed a widely inclusive clade (Aetosaurinae, *Parker, 2007*), or a genus-group taxon (*Stagonolepis sensu Heckert & Lucas, 2000*) and the tree presented by *Parker (2007)* supported the hypotheses to some extent. However, these hypotheses were quickly undermined when new cranial data were added indicating that these osteoderm characters are potentially homoplastic (*Parker, 2008b*; *Desojo, Ezcurra & Kischlat, 2012*). Indeed, in the partition analyses presented here those clades are not recovered in the endoskeletal analyses (Figs. 3C and 3D) and therefore are based almost entirely on osteoderm characters. Moreover, the full analysis shows that the main character combining these taxa (radial ornamentation of the paramedian osteoderms) is simply a plesiomorphic character of non-desmatosuchins. Thus, these data strongly suggest that even in a tree with much 'noise' (conflicting character data, weak clade support) a well-supported phylogenetic signal is coming through when all of the data are combined (*Baker & DeSalle, 1997*).

## Prospectus

Many of the discussions of dataset partitioning and character congruence and the strategies devised to deal with problems are from studies where morphological and

molecular data are being combined (e.g., *Bull et al., 1993*; *Huelsenbeck, Bull & Cunningham, 1996*; *Cunningham, 1997*; *DeSalle & Brower, 1997*; *Wiens, 1998a*, but see *Mounce, 2013*). However, there is no reason not to suspect that the same phenomena may occur in studies using purely morphological data. Different anatomical modules may possess different histories and thus present conflicting character data that can mask true phylogenetic relationships or support false ones (*Desojo, 2005*; *Parker, 2014*). Workers conducting phylogenetic analyses of morphological datasets are encouraged to explore the possibilities of incongruent subsets in their data.

Furthermore, ontogenetic change in aetosaurians is still poorly understood and it is important that specimens scored are at the same relative ontogenetic stage to rule out the possibility of differences caused by developmental history (*Taborda, Heckert & Desojo, 2015*). Determination of maturity indicators can identify synonymous taxa (originally scored separately) and provide a baseline for morphological equivalence of taxa (or specimens) used in phylogenetic studies (*Brochu, 1996*). Presently the most often used indicator for pseudosuchians, including aetosaurians) is the progression of neurocentral suture closure in the vertebral column (*Brochu, 1996*). In aetosaurians this progression begins in the caudal series and ends with the axis/atlas (*Irmis, 2007*). Unfortunately, most aetosaurian specimens lack relatively complete series (e.g., *Tecovasuchus chatterjeei*, TTU P-545) or completely lack preserved vertebrae (e.g., *Paratypothorax andressorum*, SMNS unnumbered). In others, the vertebral column is covered by the articulated carapace (*Coahomasuchus kahleorum*, NMMNH P-18496). Fortunately, other methods such as CT scanning and histological sectioning are available, but to date only a handful of specimens have been sampled and only two of these are holotypes (e.g., *Parker, Stocker & Irmis, 2008*; *Cerda & Desojo, 2011*). Aetosaur workers are encouraged to carefully determine and document maturity indicators for as many specimens as possible with a future goal of incorporating this information into phylogenetic analyses (*Taborda, Heckert & Desojo, 2015*).

As with any scientific study, this is a work in progress. Unfortunately, it is presently unclear whether phylogenetic relationships resulting from a matrix with an abundance of osteoderm characters (e.g., *Parker, 2007*) are any more correct (accurate) than those formed by a matrix with an abundance of endoskeletal (non-osteoderm) characters (this study), although I have given my reasons above for preferring the latter. This study has attempted to carefully reexamine all characters used in past analyses and to construct unambiguous characters and states. Characters were scored as carefully as possible, but certainly errors exist. The Partitioned Bremer Support analysis shows where character support for individual nodes is weak or conflicting for suites of characters and therefore can be used to examine node stability when new data are added (*Gatesy et al., 2003*; *Wahlberg & Nylin, 2003*; *Lambkin, 2004*). Thus, future analyses should look to increase the number of informative characters, fill in blanks caused by missing data and correct erroneous scorings to improve accuracy and clade support. However, they should avoid adding large numbers of poorly supported characters (i.e. heavy on missing data) just for the sake of increasing characters numbers and instead focus on creating characters that can be fully or nearly fully coded to avoid decreasing overall accuracy (*Wiens, 1998b*).

## ACKNOWLEDGEMENTS

Much of this manuscript was originally submitted for the partial requirements for the Doctor of Philosophy Degree in Geology from the University of Texas at Austin. Constructive comments on this earlier version were provided by Sterling Nesbitt, Hans Dieter-Sues, Christopher Bell, Timothy Rowe, and Julia Clarke. Access to specimens under their care was provided by T. Scott Williams and Matt Smith (PEFO); Pat Holroyd, Mark Goodwin, and Kevin Padian (UCMP); David and Janet Gillette (MNA); Julia Desojo (MACN); Jaime Powell (PVL); Ricardo Martinez (PVSJ); Sandra Chapman, Lorna Steel, and David Gower (NHMUK); Lindsay Zanno and Vince Schneider (NCSM); Sankar Chatterjee and Bill Mueller (TTUP); Matthew Carrano (USNM); Tony Fiorillo and Ron Tykoski (DMNH); Alex Downs (GR); Charles Dailey and Dick Hilton (Sierra College); Tim Rowe, Lyndon Murray, Matt Brown, and Chris Sagebiel (UT VPL). Permission to include and discuss certain unpublished specimens was provided by Jeffrey Martz, Axel Hungerbühler, Rainer Schoch, Julia Desojo, and Tomajz Sulej. Photographs of specimens were provided by Sterling Nesbitt, Julia Desojo, Jeff Wilson, Jeff Martz, Randy Irmis, and Richard Butler. Ben Creisler graciously formulated the new taxonomic name for the Petrified Forest aetosaur. Other taxonomic advice was provided by Christian Kammerer. Thorough reviews by Roland Sookias, Julia Desojo, and Andrew Farke improved the final manuscript. This is Petrified Forest National Park Paleontological Contribution 36.

### Funding

Financial assistance for this project was provided by the Jackson School of Geosciences, the Lundelius Fund, the Francis L. Whitney Endowed Presidential Scholarship, the Ronald K. DeFord Scholarship Fund, the Samuel and Doris Welles Fund, and the Systematics Association. The funders had no role in study design, data collection and analysis, decision to publish, or preparation of the manuscript.

### Competing Interests

The author declares that he has no competing interests.

### Author Contributions

- William G. Parker conceived and designed the experiments, performed the experiments, analyzed the data, contributed reagents/materials/analysis tools, wrote the paper, prepared figures and/or tables, reviewed drafts of the paper.

### New Species Registration

The following information was supplied regarding the registration of a newly described species:

Scutarx LSID: urn:lsid:zoobank.org:act:E06A8E11-5864-4717-AFA2-9021842B886D

Publication LSID: urn:lsid:zoobank.org:pub:841F81C7-A4AE-4146-94FE-DFE0A6725634

## Supplemental Information

Supplemental information for this article can be found online at http://dx.doi.org/10.7717/peerj.1583#supplemental-information.

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
