# Peer review of "Revised phylogenetic analysis of the Aetosauria (Archosauria: Pseudosuchia); assessing the effects of incongruent morphological character sets"

_PeerJ, doi:10.7717/peerj.1583_

## Round 0.1 · original submission · Minor Revisions

This is a remarkably well documented phylogenetic contribution, and one that will undoubtedly be useful to many researchers. The reviewers have raised a handful of minor issues to be cleared up prior to acceptance; these are detailed below.

GENERAL COMMENTS:
- Note the issue concerning SMNS 19003 raised by reviewer Desojo; I would advise communicating with them directly to ensure that a satisfactory solution can be reached, to avoid any potential conflicts.
- Reviewer Desojo recommends adding some additional figures for the new species, particularly for the braincase. Although I recognize that a more complete description is forthcoming, and that the purpose of this paper is largely phylogenetic, rather than centered on a single new taxon, I think that a few other details could be included, particularly to show all of the proposed autapomorphies for the new taxon.
- Reviewer Sookias (as well as reviewer Desojo) provides numerous suggestions and revisions for style and grammar as well as other minor edits; please incorporate these as applicable into the revised manuscript. These are all provided in marked-up PDF files at the PeerJ author submission portal.

OTHER EDITORIAL COMMENTS:
Abstract, last line: "which may be under different selective pressures" - should probably put this into past tense, as the evolution happened way back in the Triassic. (I have this same "bad habit" of referring to Mesozoic events in the present)
line 75: "skulls with upturned snouts, heavy armor carapaces," - change to "skulls with upturned snouts and heavy armor carapaces,"
line 152: insert "are" after "there"
line 157: add comma after "But"
line 165: "When corrected..." is a sentence fragment and should be modified
line 186: add "the" before "Parker"
line 199: add comma after "below"
line 447: add period between taxon and "Several"
lines 592-593: "These findings will be presented elsewhere." I would just remove this sentence; my personal preference is to avoid any promise of future papers so as to avoid potential confusion down the line in case plans change.
line 1107: "SMSN" should be "SMNS" - there are a few other instances of this transposition elsewhere that should be cleaned up, too.

- For Figure 2, you might mention that those are dorsal silhouettes. Also, because it is indicated as a modified figure from another source, please confirm that it does not require any additional permissions to be released in CC-BY (I don't have the original handy, so I cannot speak to how much it has been modified or redrawn).
- For Figure 4, the line drawing might be improved with some shading for the various fenestrae and foramina, to help the image "pop". In its current incarnation, the various depths on the specimen somewhat run into each other. At a minimum, if the supratemporal fenestrae are given a gray or black fill, it could be a big help.
- The various A/B figures should be given individual numbers, rather than letter/number combinations.
- Figure B14 needs letters for each part.

·

Basic reporting

Fits the bill here.

Experimental design

Again here.

Validity of the findings

Again here.

Additional comments

This is an extremely thorough piece of work, well worthy of publication in PeerJ, and I look forward to seeing it out.

There are a number of very minor revisions which I would like the author to carry out. These are marked up in the PDF. They are largely language issues, although there is an issue with formulation of one of the phylogenetic characters.

Beyond what is specifically marked up, in several places the article is rather wordy, and I feel many of the more theoretical sentences in the earlier parts of the paper in particular could be substantially condensed. This would aid reading the paper and make the underlying messages clearer.

·

Basic reporting

It is a very important paper for the Pseudosuchian archosaurs, particularly for Aetosauria. The author incorporate new characters, discuss each taxon, described a new species, and discuss incongruent data set (osteoderm, skull, postcranial character conflict). I am very happy with this phylogeny update.
I incorporated my comments and suggestions on the pdf file, and strongly recommended its publication. However, few crucial points to be discuss and clarify:
. The SMNS 19003 is a material understudy by Schoch and Desojo (see Desojo and Schoch 2014 SVP abstract) and submitted. As I described in the pdf, the author was one of the reviewer for the paper about SMNNS 19003 submitted last November 2014, and probably he thought that the paper is in press, but unfortunately, it was rejected. So, we are submit it again. So, the picture and score are overlap with our paper.
- Several of the new characters are base on published and unpublished PhD Thesis (e.g. Desojo 2005), so, should be clarify at the beginning of the analysis and each description character (see pdf comments).
- The new species described should be figured in detail, more the braincase characters in lateral and ventral view.

There are some update of the references taht should be incorporate (see pdf)

Experimental design

the phylogeny methodology is accord to the actual phylogeny analysis

Validity of the findings

the same to the editor

Additional comments

The same to the editor

---

## Round 0.2 · accepted · Accept

Thank you for your close attention to the reviewer and editor comments for the previous version. The manuscript is ready to move forward, in my opinion.

---

## Author Rebuttal · Round 0.2

Dear Bill,

I have to say that I have never made such an experience before. [Figure] it will appear in January 2016 in another journal.

In our revision we considered all valuable points of the reviews, so they were certainly helpful.

I have no objections against you using the photos and referring to the characters.

Best wishes
Rainer

[Quoted text hidden]
--
Dr. Rainer Schoch
Staatliches Museum fuer Naturkunde